# Genetically stable poliovirus vectors activate dendritic cells and prime antitumor CD8 T cell immunity

Mubeen M. Mosaheb[1], Elena Y. Dobrikova[2], Michael C. Brown[2], Yuanfan Yang [3], Jana Cable[1], Hideho Okada[4,5], Smita K. Nair[6], Darell D. Bigner[2], David M. Ashley[2] & Matthias Gromeier[1,2]*

Viruses naturally engage innate immunity, induce antigen presentation, and mediate CD8 T cell priming against foreign antigens. Polioviruses can provide a context optimal for generating antigen-specific CD8 T cells, as they have natural tropism for dendritic cells, preeminent inducers of CD8 T cell immunity; elicit Th1-promoting inflammation; and lack interference with innate or adaptive immunity. However, notorious genetic instability and underlying neuropathogenicity has hampered poliovirus-based vector applications. Here we devised a strategy based on the polio:rhinovirus chimera PVSRIPO, devoid of viral neuro-pathogenicity after intracerebral inoculation in human subjects, for stable expression of exogenous antigens. PVSRIPO vectors infect, activate, and induce epitope presentation in DCs in vitro; they recruit and activate DCs with Th1-dominant cytokine profiles at the injection site in vivo. They efficiently prime tumor antigen-specific CD8 T cells in vivo, induce CD8 T cell migration to the tumor site, delay tumor growth and enhance survival in murine tumor models.

[1] Department of Molecular Genetics & Microbiology, Duke University Medical School, Durham, NC 27701, USA. [2] Department of Neurosurgery, Duke University Medical School, Durham, NC 27701, USA. [3] Department of Pathology, Duke University Medical School, Durham, NC 27701, USA. [4] Parker Institute for Cancer Immunotherapy, University of California at San Francisco, San Francisco, CA 94129, USA. [5] Department of Neurological Surgery, University of California at San Francisco, San Francisco, CA 94129, USA. [6] Department of Surgery, Duke University Medical School, Durham, NC 27701, USA. *email: grome001@mc.duke.edu

Bolstering the potency or frequency of antitumor cytotoxic T lymphocyte (CTL) responses has achieved durable responses in cancer patients[1–5]. Generating such responses in situ requires an appropriate costimulatory context coinciding with presentation of target antigen by MHC class I. Viral vectors offer a compelling approach to engage CTLs by providing intracellular expression of both, pathogen-associated molecular patterns and the desired antigen. This recapitulates the cues CTLs have evolved to respond to upon intracellular pathogen infection; in essence portraying the target antigen as a viral signature.

A clinically feasible poliovirus vector platform has remained elusive, despite obvious advantages: tropism for antigen-presenting cells[6,7], a rapid lifecycle capable of propagating vector-encoded antigen, no interference with immune activation and antigen presentation, and robust engagement of inflammatory responses. We report the engineering of genetically stable vectors based on the highly attenuated PVSRIPO [type 1 poliovirus (Sabin) live-attenuated vaccine containing a rhinovirus type 2 internal ribosomal entry site (IRES)][8]. PVSRIPO has inherent tropism for macrophages and DCs in primates[6], and exhibits an unusual phenotype in such cells, characterized by lingering, sublethal infection and profound proinflammatory stimulation[9]. PVSRIPO is devoid of neuropathogenicity, even after high-dose intracerebral inoculation in non-human primates[10] and in human subjects with recurrent WHO grade IV malignant glioma (glioblastoma; GBM)[11]. Intratumoral delivery of PVSRIPO has achieved clinical and radiographic responses with durable long-term survival (>36 months) in 21% of patients with recurrent GBM, an indication notorious for lacking immune engagement[11].

Unlike the extremely heterogeneous adult malignant gliomas, the predominantly pediatric Diffuse Midline Gliomas (DMG) frequently are defined by a loss in histone 3 (H3K27) trimethylation[12,13]. Approximately 80% of DMGs of the pons [Diffuse Intrinsic Pontine Glioma (DIPG)] express H3.3$^{K27M}$, which also occurs in spinal and thalamic DMGs. In addition, H3.3$^{G34R/V}$ is associated with hemispheric high grade gliomas in pediatric patients[12]. Loss of H3K27 trimethylation has been implicated in aberrant gene expression control and tumorigenesis[14].

H3.3$^{K27M}$ is a high affinity HLA-A2-restricted tumor neoantigen and T cells bearing the H3.3$^{K27M}$-specific TCR lysed H3.3$^{K27M}$-positive glioma cells in vitro and in vivo[15]. Devising effective immunotherapies for DMG/DIPG is an urgent mandate, as there is no standard treatment for this invariably fatal disease with dismal prognosis. We describe a genetically stable PVSRIPO-based vector platform to generate antitumor responses against the H3.3$^{K27M}$ epitope. We show that PVSRIPO-based vectors target DCs to: elicit sublethal viral translation and propagation resulting in expression of foreign epitopes in a highly adjuvated context; induce DC maturation markers; provoke type I/III interferon (IFN) release; present the H3.3$^{K27M}$ epitope to T cells; generate locoregional proinflammatory activation, immune cell infiltration and DC activation in vivo; and trigger DC migration to immunization site-draining lymph nodes in vivo, where they express the H3.3$^{K27M}$ epitope. Vector immunization induces CD8 T cell infiltration into tumors, and exhibits significant anti-tumor efficacy in immunocompetent rodent tumor models.

## Results

**Genetically stable PVSRIPO vectors.** Among diverse proposals for poliovirus-derived vectors, "polyprotein fusion" approaches are most advanced[16–20]. Enterovirus + strand RNA genomes encode a single polyprotein, which is processed by two viral proteases (2A$^{pro}$, 3C$^{pro}$). For polyprotein fusion, foreign polypeptides are N-terminally or internally fused to the polyprotein and released via an engineered viral protease cleavage site (Fig. 1a, b)[16]. Poliovirus polyprotein fusion vectors delivering simian immunodeficiency virus (SIV)-derived polypeptides or the ovalbumin (OVA) model antigen were tested in non-human primates[16–18] and in human CD155-transgenic (hCD155-tg) mice[20]. Effective humoral, CD4- and CD8 T cell responses[16–20] protected macaques against lethal SIV challenge[18]. Yet, this vector design, where burdensome foreign sequences were simply added to the viral genome, triggered rapid deletion events[21]. We reasoned that to achieve genetic stability, foreign inserts must be functionally integrated into the genome, in order to contribute to viral fitness. All enterovirus genomes contain a cryptic AUG at the base of stem loop domain (SLD) VI of the IRES (Fig. 1a)[22]. Placing the cryptic AUG into KOZAK context and deleting SLD VI yields viable virus with growth deficits[23,24]. We used this scenario to incentivize insert retention, by putting the cryptic AUG into KOZAK context and replacing SLD VI with a coding region for foreign polypeptides (Fig. 1b, g).

We constructed a vector for the OVA class I epitope (SIINFEKL) along these guidelines (mOVA1; Fig. 1b), using a mouse-adapted PVSRIPO variant (mRIPO)[9] as backbone (to enable proof-of-principle immunization tests in mice). We assessed genetic stability through 20 serial passages in HeLa cells. Corroborating earlier findings[23], the OVA insert was retained over 20 passages. A single nt adaption emerged in passage 10: mixed C609/A that became A609 by passage 20 (Fig. 1d), suggesting a fitness advantage conveyed by A609. C609A is a missense mutation (L/M; Fig. 1d) that does not affect SIINFEKL epitope processing and presentation. Cloned mOVA2 carrying the A609 substitution remained genetically intact through 20 serial passages in HeLa cells (Fig. 1e); mOVA2 formed larger plaques with enhanced growth compared to mOVA1 (Fig. 1f).

A vector delivering the DMG/DIPG H3.3$^{K27M}$ epitope [RIPO (H3.3); Fig. 1g] remained unchanged after 20 serial passages in HeLa cells (Supplementary Fig. 1). RIPO(H3.3)'s plaque size and growth kinetics were enhanced compared to PVSRIPO with SLD VI deleted [RIPOδ6; Fig. 1h], suggesting that the foreign insert conveys a fitness advantage. RIPO(H3.3) retained the neuron-specific translation deficit of its PVSRIPO parent, which defines neuroattenuation in vivo[25,26]. This was evident as deficient RIPO(H3.3) growth in HEK293 cells (adrenal precursor cells of neuroectodermal origin[27]) (Supplementary Fig. 2). HEK293 cells accurately reflect the neuroattenuation phenotypes of PVSRIPO and the poliovirus (Sabin) vaccines[28,29]. The PVSRIPO platform can be flexibly adapted to accommodate diverse inserts; we derived genetically stable vectors expressing peptides encoding the isocitrate dehydrogenase 2 (IDH2) R172G mutation, or SIV p55 Gag-derived peptides (Supplementary Fig. 3).

**mOVA2 initiates translation from the intended AUG.** The C609A substitution in mOVA2 resulted in tandem, in-frame AUGs separated by 18nt; both of these were in ideal KOZAK context (A/G . . **AUG** G) (Supplementary Fig. 4a). RIPO(H3.3) inherently contains tandem, in-frame AUGs (separated by 36nt); the downstream AUG, however, is in poor KOZAK context (C . . **AUG** A) (Fig. 1g). mOVA2 acquiring a new in-frame AUG in optimal KOZAK context raises questions about initiation codon usage in our vector design, as usage of a downstream AUG would truncate the foreign coding region. To examine initiation codon usage in our vectors, we used mOVA2 variants with the upstream AUG in poor KOZAK context and/or a frameshift in-between the two tandem AUGs. These tests showed that the intended

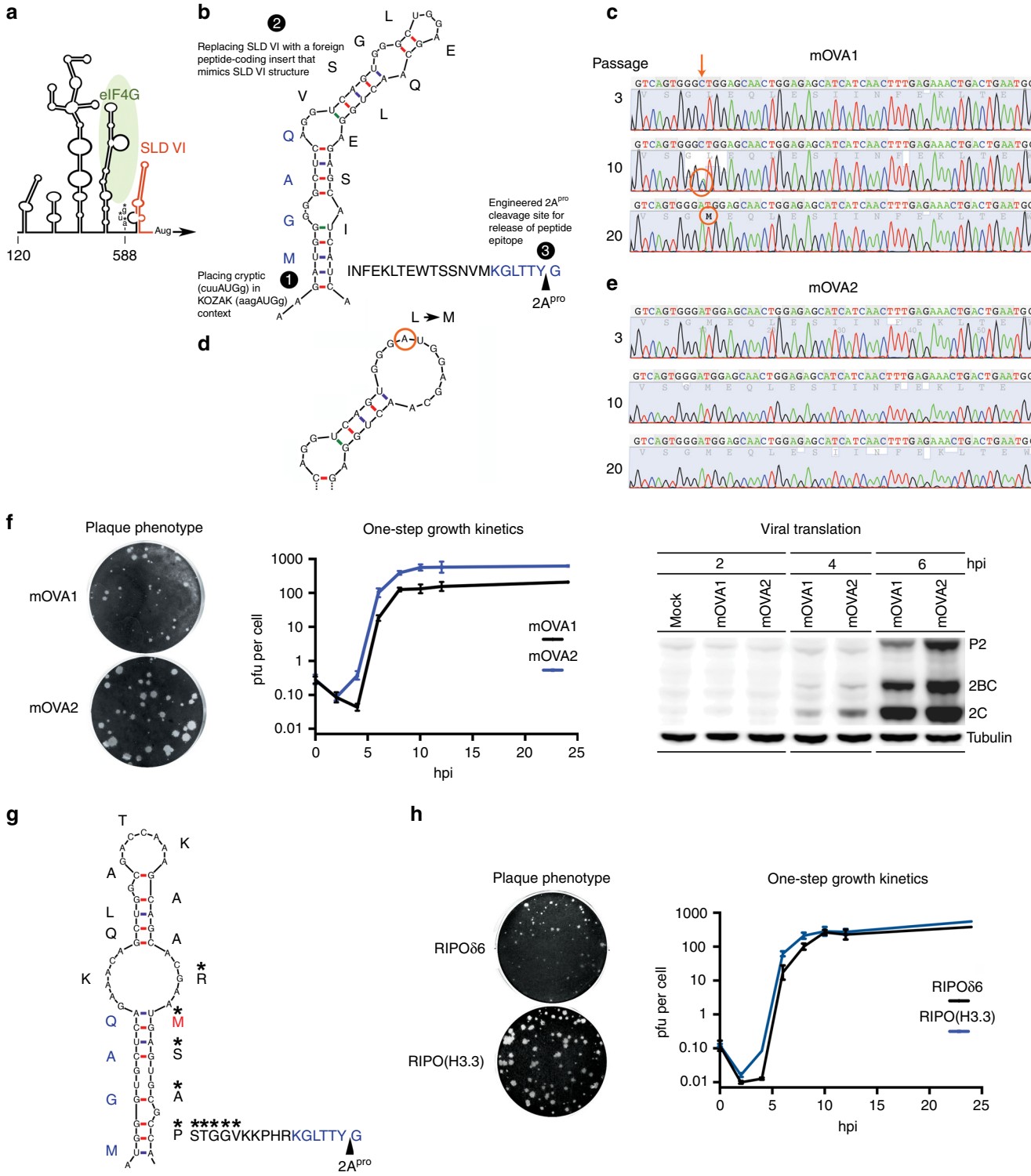

initiation AUG at nt 588 in optimal KOZAK context is preferred (Supplementary Fig. 4).

**mOVA2 and RIPO(H3.3) express and process foreign epitopes.** With our vector design, the H3.3$^{K27M}$/OVA epitopes are fused to the viral P1 or VP0 precursors and processed by 2A$^{pro}$ cleavage (Fig. 2a). An H3.3$^{K27M}$-specific antibody probe detected such H3.3$^{K27M}$-fusion polypeptides in RIPO(H3.3)-infected HeLa cells (Fig. 2b). These appeared early (H3.3$^{K27M}$-P1 peaked at 3hpi) and declined thereafter, presumably due to viral proteolytic

processing. Processing of the H3.3$^{K27M}$-VP0 fusion product was delayed compared to H3.3$^{K27M}$-P1, in line with the poliovirus processing sequence (Fig. 2b)[30]. See Supplementary Fig. 5 for viral fusion-polypeptide expression analyses in mOVA2-infected HeLa cells. Expression of viral H3.3$^{K27M}$/OVA-fusion proteins with our vector design corroborates our studies of initiation codon usage (Supplementary Fig. 4).

**Sublethal infection of hCD155-tg mouse BMDCs with mOVA2.** Monocytes, macrophages, and DCs express the poliovirus receptor

**Fig. 1 PVSRIPO-based polyprotein fusion vectors with IRES SLD VI replacement. a** An engineered type 1 IRES with the conserved, cryptic AUG initiating a foreign ORF mimicking the predicted overall structure of the HRV IRES SLD VI (orange). The approximate 'footprint' of eIF4G in type 1 IRESs[62,63], anchoring the eIF4G:4A:4B translation initiation helicase, is indicated. **b** Genetic structure of mOVA1, expressing a VSGLEQLE**SIINFEKL**TEWTSSNVM polypeptide, flanked by a (viral) N-terminal MGAQ sequence and a C-terminal engineered cleavage site (KGLTTY^G) for poliovirus 2A[pro] (blue letters). Salient vector design principles are indicated: [1] convert the cognate cryptic AUG into the initiation AUG; [2] replace SLD VI with an epitope-encoding sequence mimicking SLD VI; [3] engineer a 2A[pro] cleavage site for epitope release. **c**, **d** Sequence of the mOVA1 insert (nt 600-657) at passages 3, 10 and 20: a C-to-A substitution at nt 609 emerging in passage 10 (**c** circled) gives rise to a Leu-to-Met coding change in mOVA2 (**d**). **e** Sequence chromatogram of mOVA2 (nt 600-657) at passages 3, 10 and 20. **f** Plaque phenotypes (left panel), one-step growth curves in HeLa cells (middle panel) and translation kinetics in HeLa cells (right panel) of mOVA1 and 2. **g** Genetic structure of RIPO(H3.3) expressing a KQLATKAA**RM*SAPSTGGV**KKPHR polypeptide derived of histone 3.3; the Lys-to-Met substitution at aa27 is indicated in red. The previously defined neoepitope in mutant H3.3[K27M][15] is indicated by asterisks. **h** Plaque phenotypes (left panel) and one-step growth curves in HeLa cells (right panel) of RIPOδ6 and RIPO(H3.3). **f**, **h** n = 3 independent samples for 0–12hpi; error bars denote Standard Error of the Mean (SEM).

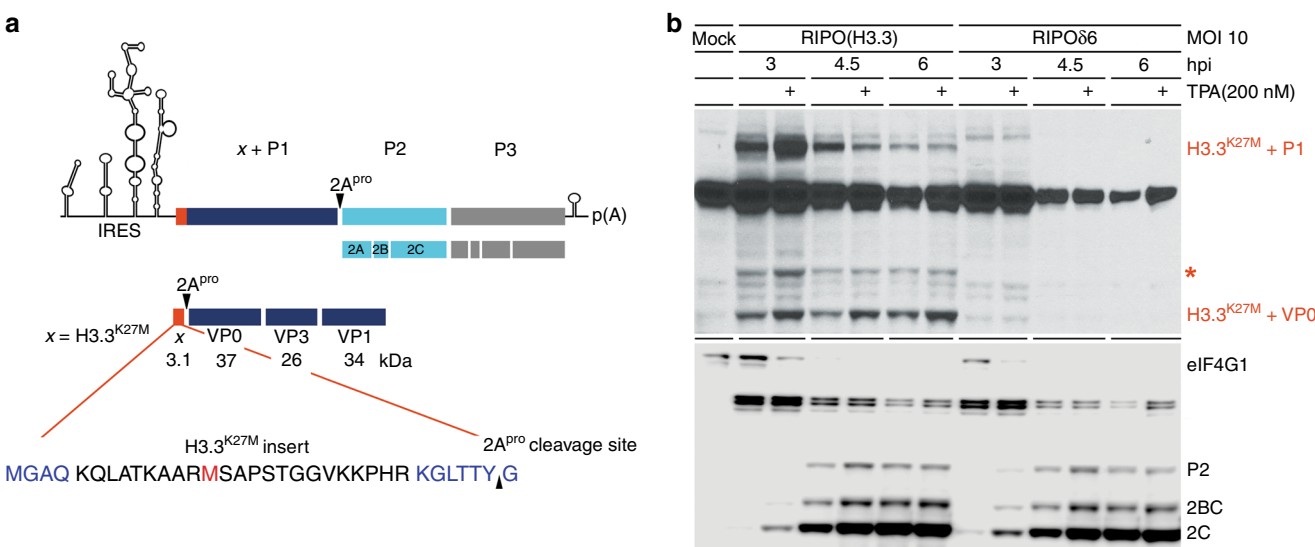

**Fig. 2 RIPO(H3.3) expresses and processes the foreign H3.3[K27M] antigen. a** Processing map for RIPO(H3.3); the H3.3[K27M] epitope is shown in red. **b** Top panel. Viral P1 and VP0 precursor polypeptides fused to the H3.3[K27M] epitope were detected. We observed an additional, weaker band in RIPO (H3.3)-infected cells that cannot be assigned to any of the predicted viral precursor polypeptides in size (*). Since the signal strength of this band over time recapitulated the pattern of H3.3[K27M]-P1, we assume that this band represents a degradation product of H3.3[K27M]-P1. Bottom Panel. eIF4G1 cleavage, viral P2/2BC/2C expression in HeLa cells infected with RIPO(H3.3). 12-O-Tetradecanoylphorbol-13-acetate (TPA) was used to stimulate viral translation[64].

CD155[31] and are natural, high priority poliovirus targets in primates[6]. Accordingly, poliovirus infects monocytes, macrophages, and DCs in vitro[7,31,32]. In line with productive infection in primate CD11c+ macrophages/DCs in vivo[6], poliovirus infection yields cytopathogenic virus replication and host protein synthesis shut-off in human DCs derived from PBMCs[7]. In contrast, PVSRIPO exhibits a sublethal phenotype in human DCs[9], with protracted, low-level viral translation and propagation, accompanied by sustained, potent type I-dominant IFN activation[9].

We tested mOVA2-infected DCs for: viral translation and cytotoxicity; proinflammatory cytokine production; expression and presentation of OVA class I epitope; and antigen-specific activation of CD8+ cytotoxic T cells. To this end, we established bone marrow-derived DC (BMDC) cultures from hCD155-tg mice either differentiated with GMCSF/IL4 (GMCSF-BMDCs; Fig. 3a) or with Fms-related tyrosine kinase 3 ligand (FLT3L-BMDCs; Fig. 3b). The latter consist of a mixture of CD8α+ (Batf3+ DCs) and CD8α− DCs[33,34]; CD8α+ DCs have a prominent role in eliciting CTL responses[35,36]. GMCSF− (Fig. 3a) or FLT3L− (Fig. 3b) BMDCs were either mock-infected, or infected with mOVA2 at a multiplicity of infection (MOI) of 10. As a positive control for proinflammatory stimulation, we used lipopolysaccharide (LPS) (Fig. 3a, b).

mOVA2 infection of GMCSF- and FLT3L-BMDCs revealed the absence of eIF4G1 cleavage, the hallmark of poliovirus cytopathogenicity[37]. We did not detect productive viral translation, i.e. viral 2C/2BC, in GMCSF-BMDCs (Fig. 3a). In contrast, viral protein synthesis was abundant in FLT3L-BMDCs (Fig. 3b). Viral translation peaked early (12hpi) and gradually declined without evidence for cytopathogenicity (i.e. eIF4G1 cleavage), indicating an ability of the FLT3L-BMDCs to prevent cytotoxic damage stemming from viral translation (Fig. 3b).

**mOVA2-induced type I IFN-dominant activation of mouse BMDCs.** Both, GMCSF- and FLT3L-BMDCs responded to mOVA2 infection with type I IFN activation signatures, e.g. induction of p-STAT1(Y701), STAT1, TAP1 and ISG15 (Fig. 3a, b). This response was delayed in FLT3L- relative to GMCSF-BMDCs, especially at 12hpi (compare mOVA2-infected to LPS-treated cultures; Fig. 3a, b). Results of repeat assays are shown in Supplementary Fig. 6a, b. mOVA2 infection led to the upregulation of DC maturation markers (CD40/80/86) and PD-L1 in a pattern that was qualitatively similar, but quantitatively diverging in FLT3L- vs. GMCSF-BMDCs (Fig. 3a, b). Levels of MHC I and -II remained unchanged in mOVA2-infected GMCSF- and FLT3L-BMDCs throughout the assay (Supplementary Fig. 6a, b). Delayed type I IFN responses in FLT3L-BMDCs were also evident

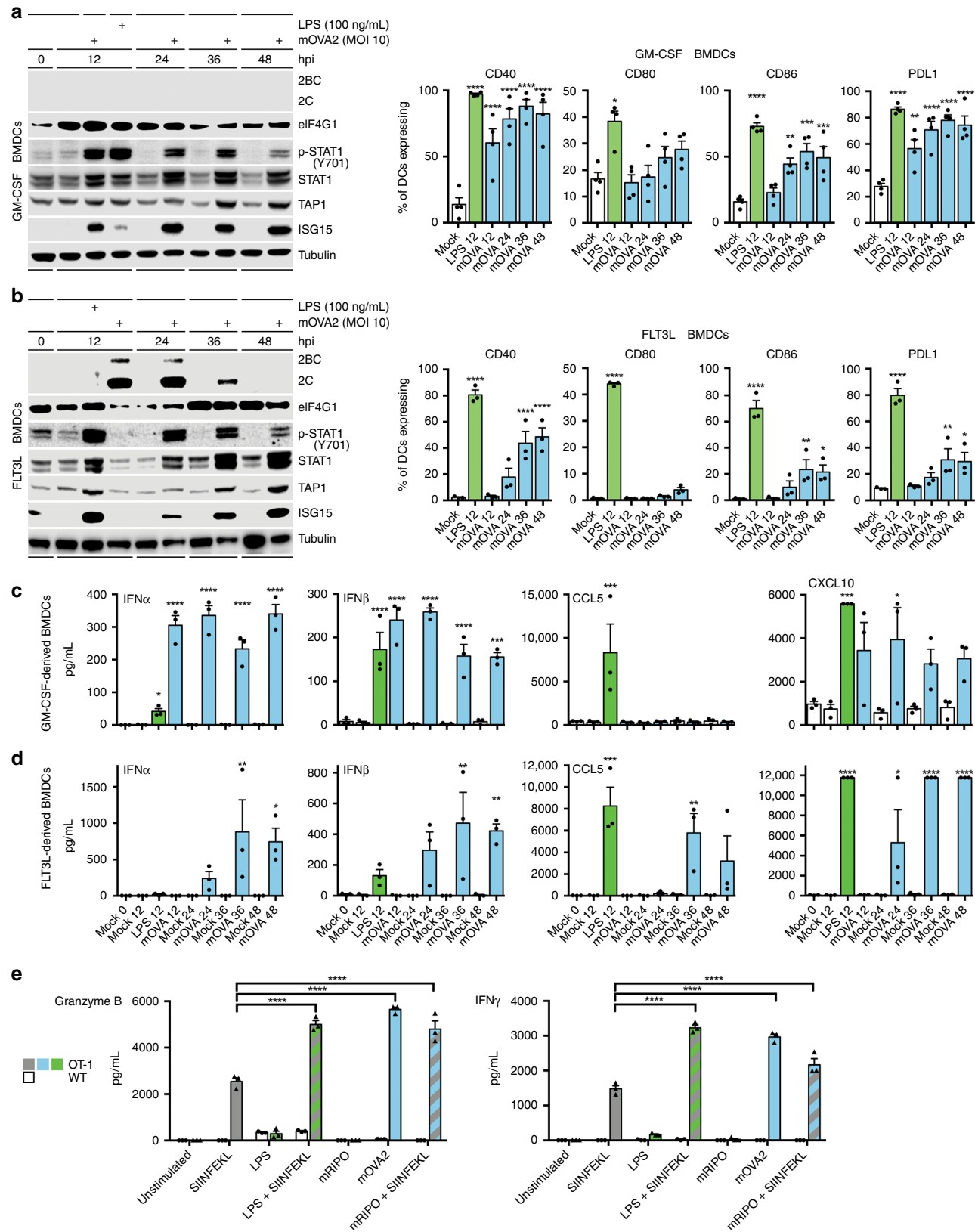

in cytokine release assays in mOVA2-infected GMCSF- (Fig. 3c) and FLT3L-BMDCs (Fig. 3d). In both, GMCSF- and FLT3L-BMDCs, substantial responses were recorded for type 1 IFNs (IFNα/β), CCL5 (RANTES) and CXCL10 (Fig. 3c, d) (other tested cytokines had low or absent responses; Supplementary Fig. 7a, b).

While proinflammatory cytokine responses in FLT3L-BMDCs were delayed, their amplitude was higher compared to GMCSF-BMDCs (Fig. 3c, d). Collectively, these findings suggest that mOVA2 infection of BMDCs stimulated their antigen-presenting and T cell co-stimulation capacity.

**Fig. 3 mOVA2 activates DCs and mediates SIINFEKL epitope presentation to CD8 T cells. a, b** Immunoblot and flow cytometry analysis of GMCSF- (**a**) and FLT3L- (**b**) BMDCs showing type I IFN-dominant activation and upregulation of costimulatory molecules upon infection with mOVA2. For co-stimulatory molecule expression, asterisks denote statistically significant differences from mock (0 h)-treated samples as determined by Two-way RM-ANOVA protected Dunnett's post hoc test; (**a**) $n = 4$; (**b**) $n = 3$. Repeat immunoblot assays and gating strategy are shown in Supplementary Fig. 6a–c, respectively. **c, d** Multiplex ELISA showing cytokine profiles of infected GMCSF- (**c**) and FLT3L- (**d**) BMDCs. Asterisks denote statistically significant differences from mock (0 h)-treated samples determined by One-way ANOVA protected Dunnett's post hoc test; $n = 3$. **e** GMCSF-BMDC/OT-I co-culture assays. Asterisks denote statistically significant differences determined by Two-way ANOVA protected Sidak's multiple comparison test, $n = 3$. The assay was performed three times; representative results are shown. **a–e** Error bars denote SEM. $*P < 0.05$, $**P < 0.01$, $***P < 0.001$, $****P < 0.0001$; $n =$ number of independent replicates.

**mOVA2-infected BMDCs present SIINFEKL epitope**. To test if mOVA2-infected and activated DCs present the SIINFEKL epitope and stimulate T cell function, we infected GMCSF-BMDCs with mOVA2 (MOI 10) and co-cultured them with either OT-I (SIINFEKL-specific) or naïve CD8$^+$ T cells. We compared IFN-γ and granzyme B production by mOVA2-infected- to unstimulated-, SIINFEKL peptide-, LPS-, LPS + SIINFEKL−, mRIPO- or mRIPO + SIINFEKL-stimulated GMCSF-BMDCs (Fig. 3e). mOVA2 infection alone mirrored the response to LPS + SIINFEKL-stimulation (Fig. 3e). mRIPO was devoid of activity in this assay, but mRIPO + SIINFEKL-stimulation instigated responses that exceeded those to SIINFEKL-stimulation alone (Fig. 3e). These observations indicate expression, processing, and presentation of foreign polypeptides by PVSRIPO-based vectors in DCs, and illustrate the costimulatory potential of PVSRIPO-based vector infection.

**Adjuvant effects of mOVA2 in vivo**. A salient argument for viral vector-delivery of antigens is local adjuvancy from the stimuli provided by virus challenge. To test this, we immunized hCD155-tg mice with either mOVA2 or DMEM (Mock) by the intramuscular (i.m.) route and isolated skeletal muscle tissue from the inoculation site to assess the presence of mOVA2 by plaque assay; isolated skeletal muscle tissue from the inoculation site (24hpi) for tests of cytokine responses (Fig. 4a; compare to in vitro BMDC infection in Fig. 3c, d; Supplementary Fig. 7); assessed local immune cell infiltrates by flow cytometry (Fig. 4b); and tested DC maturation status by evaluating CD40 and CD86 expression on local infiltrating DCs (Fig. 4c).

A breadth of cytokines was induced by mOVA2 infection in vivo (Fig. 4a). The four principal responses to mOVA2 infection of DCs in vitro, IFNα/β, CCL5, CXCL10 (Fig. 3c, d), and IFNγ evident in the OT-I assay (Fig. 3e), also occurred in vivo (Fig. 4a). As suggested by the pattern of the local cytokine/chemokine signature, there was profuse recruitment of immune cells, e.g. DCs, macrophages and T cells, to the site of immunization (Fig. 4b). Confirming data with in vitro infection of BMDCs with mOVA2 (Fig. 3a, b), local DCs acquired maturation markers (CD40 and CD86), with CD40 being the most responsive (Fig. 4c).

Locoregional inflammation elicited by i.m. mRIPO-vector administration, immune cell infiltration, and costimulatory DC activation indicated that local DC infection with vector, and (infected) DC migration to draining lymph nodes may occur in vivo. Therefore, we examined antigen expression in (popliteal) lymph nodes draining the vector inoculation site in vivo. To this end, we used the mouse-adapted form of RIPO (H3.3) [mRIPO(H3.3)] for i.m. immunization of hCD155-tg mice, because a suitable antibody for immunohistochemistry (IHC) of the H3.3$^{K27M}$ epitope is available. H3.3$^{K27M}$ IHC revealed that mRIPO(H3.3) had drained to popliteal lymph nodes in H3.3$^{K27M}$-expressing cells, possibly migrated mRIPO (H3.3)-infected DCs (Fig. 4d).

As reported previously for wt poliovirus after i.m. inoculation in hCD155-tg mice[38], the extent of mRIPO(H3.3) propagation in skeletal muscle at the inoculation site was very modest (Supplementary Fig. 12). Sequencing of mRIPO(H3.3) virus isolated from skeletal muscle tissue at the immunization site (at day 4 after i.m. immunization) revealed that the foreign insert remained genetically intact (Supplementary Fig. 13). This corroborated our data on genetic stability during in vitro passaging (Supplementary Fig. 1).

**Vector immunization primes CD8 T cells in vivo**. We investigated mOVA2′s capacity for generating antigen-specific CD8 T cell responses in vivo. hCD155-tg mice were primed on day 0 and boosted on day 14 with mRIPOδ6 (empty vector), poly(I:C), poly (I:C) + SIINFEKL peptide, mRIPOδ6 + SIINFEKL or mOVA2 by i.m. inoculation. SIINFEKL-specific CD8 T cell responses were monitored by IFN-γ ELISPOT with splenocytes (Fig. 5a, b) and pentamer staining of peripheral blood (Fig. 5c, d). Both assays revealed mOVA2-generated SIINFEKL-specific CD8 T cell responses in vivo (Fig. 5a, d). Regimens lacking the antigen [mRIPOδ6, poly(I:C)] did not yield responses. The response to mOVA2 immunization was consistently superior to poly(I:C) + SIINFEKL or mRIPOδ6 + SIINFEKL both by ELISPOT and pentamer staining (Fig. 5a, d). This suggests that the range of adjuvancy provided by the vector; and vector-mediated expression of the epitope inside vector-activated cells contribute to an efficient effector immune response in vivo. If mOVA2′s capacity for eliciting SIINFEKL-specific CD8 T cell responses is due to infection of-, epitope expression in-, and costimulatory activation of DCs, then adoptive transfer of wt C57Bl6 mice (which do not support poliovirus infection) with ex vivo mOVA2-infected syngeneic FLT3L BMDCs from hCD155-tg mice should have a similar effect. This was indeed the case (Supplementary Fig. 16).

We also tested the capacity of mRIPO(H3.3) to elicit H3.3$^{K27M}$-specific CD8 T cell responses in vivo. We first generated a suitable mouse host, since the H3.3$^{K27M}$ epitope is HLA-A2 restricted[15]. The offspring of crossing (homozygous) hCD155-tg mice with (homozygous) AAD-tg C57Bl6 mice[39] (see below) was used for mRIPO(H3.3) immunization as described for mOVA2 above. For controls, we immunized litter mates with Complete Freud's Adjuvant (CFA) and H3.3$^{K27M}$ peptide (RMSAPSTGGV) emulsion (Fig. 5e, f). Staining of peripheral blood performed with a HLA-A2-H3.3$^{K27M}$ tetramer demonstrated that immunization with mRIPO(H3.3) generated H3.3$^{K27M}$ specific CD8 T cells (Fig. 5e, f).

**Anti-tumor efficacy of mOVA2 Immunization**. We first evaluated our PVSRIPO vectors in a standard, spontaneous, and relatively 'non-immune engaged'[40] immunocompetent mouse tumor model, B16F10.9-OVA melanoma. The OVA class I epitope SIINFEKL strongly binds to H2Kb in C57Bl6 mice, permitting evaluation of mOVA2 without MHC mismatch problems.

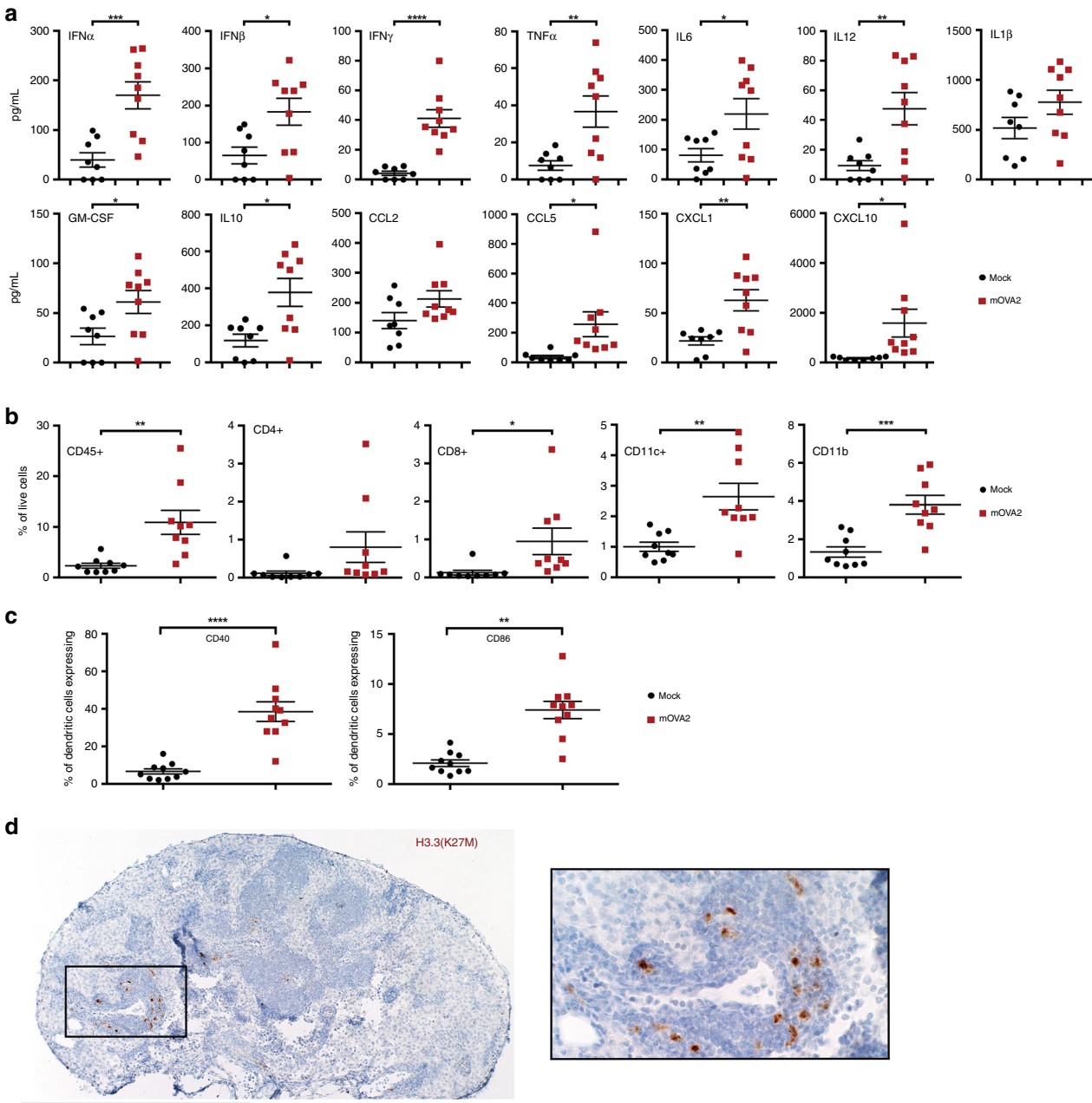

**Fig. 4 Intramuscular immunization with PVSRIPO vectors induced locoregional inflammation. a** Cytokine induction profile in gastrocnemius muscle 12 h post i.m. immunization with mOVA2 ($n = 9$) or DMEM (Mock; $n = 8$). **b** Immune infiltrates in treated muscle 24 h post immunization with mOVA2 ($n = 9$) or DMEM (Mock; $n = 9$). **c** Upregulation of costimulatory molecules on muscle-infiltrating DCs upon immunization mOVA2 ($n = 10$) or DMEM (Mock; $n = 10$). **b**, **c** Representative gating is shown in Supplementary Figs. 8, 9. **d** Immunohistochemistry of the H3.3$^{K27M}$ epitope in a popliteal lymph node draining the mRIPO(H3.3) inoculation site. H3.3$^{K27M}$ positive cells, presumably antigen-presenting cells infected with mRIPO(H3.3), are observed. H3.3$^{K27M}$ staining in sections from lymph nodes of a mock-immunized animal, and isotype-matched negative control staining of sections from the same lymph node are shown in Supplementary Figs. 10, 11, respectively. **a–c** Experiments were repeated twice, representative results are shown. Asterisks denote statistical significance determined by two-tailed student's $t$ test. Error bars denote SEM. *$P < 0.05$, **$P < 0.01$, ***$P < 0.001$, ****$P < 0.0001$; $n =$ number of independent animals.

Therefore, to assess if mOVA2-instigated CD8 T cell responses exert anti-tumor efficacy, we challenged the B16F10.9-OVA tumor model in hCD155-tg mice. In mOVA2-immunized mice, tumor progress was significantly slowed with a significant survival benefit compared to mRIPOδ6-immunized mice (Fig. 6a, b). To mechanistically correlate these findings with mOVA2's capacity to induce tumor antigen-specific CD8 T cell responses (Fig. 5a–d), we studied B16F10.9-OVA tumor immune infiltrates (Fig. 6c).

This is not feasible with the approach described for Fig. 6a, b, because the mOVA2-induced treatment effect reduced tumor size at the testing interval (day 11), preventing comparison of mRIPOδ6- vs. mOVA2-immunized animals in similarly sized tumors (Fig. 6a). Therefore, B16F10.9-OVA tumors were implanted (day 0) followed by i.m. immunization with mRIPOδ6 or mOVA2 (day 1) and tumors were isolated at day 11 for analyses of CD8 and CD4 T cell infiltration by flow cytometry (Fig. 6c;

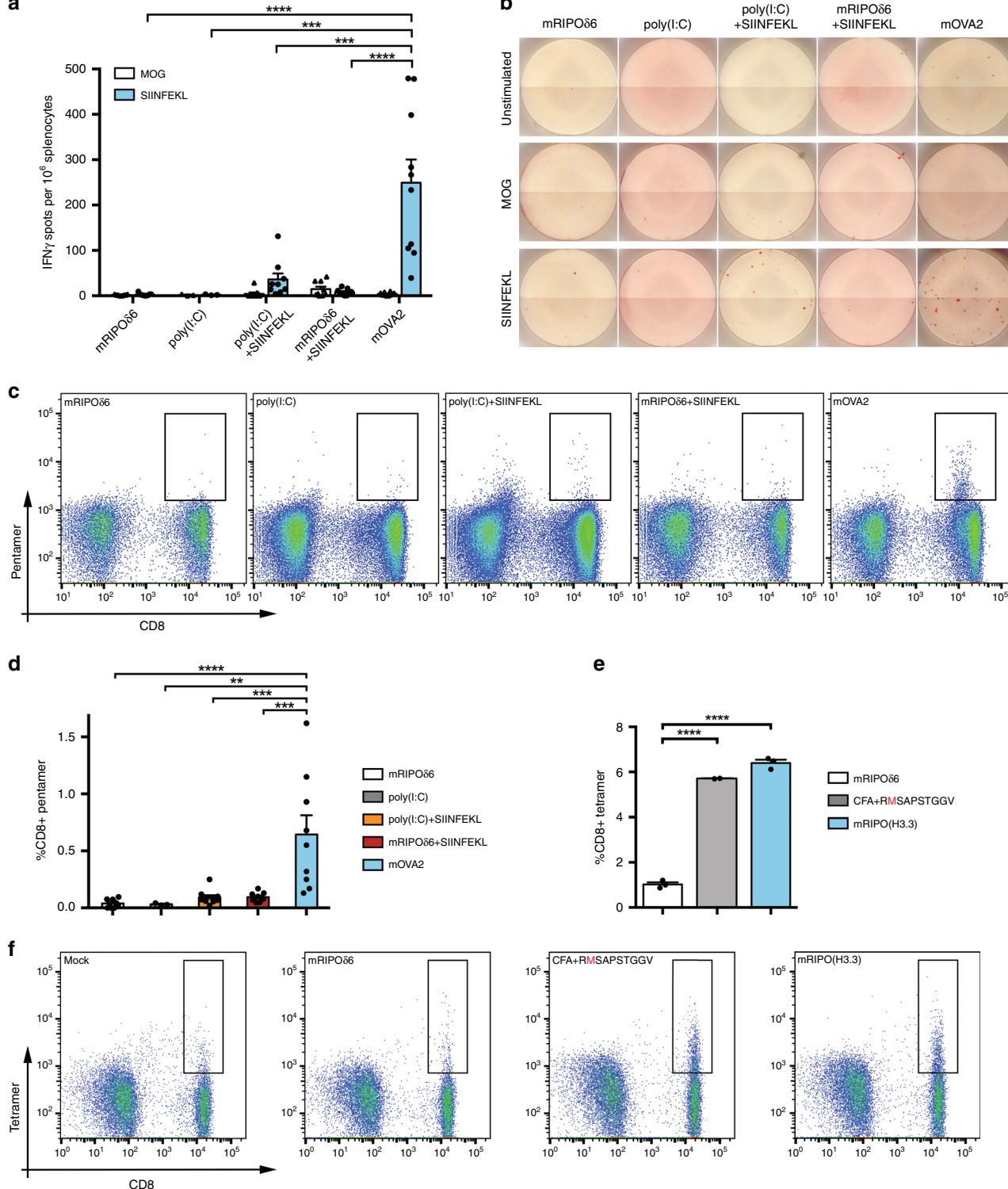

Supplementary Fig. 17). mOVA2-immunized mice had significantly higher T cell (CD3+) infiltration in the tumor, when compared to mRIPOδ6-immunized mice (Fig. 6c). Also, the CD8:CD4 T cell ratio was significantly skewed towards CD8 T cells in mOVA2-immunized mice (Fig. 6c). This suggests that SIINFEKL-specific CD8 T cells generated by mOVA2 immunization migrate to and infiltrate the tumor and mediate the decrease in tumor burden.

Next, we assessed if the therapy effect of mOVA2 immunization: correlates with the enhanced frequency of SIINFEKL-specific CD8

T cells [compared to poly(I:C) + SIINFEKL; Fig. 5a–c]; and depends on the presence of the OVA antigen in the tumor target. To this end, hCD155-tg mice were immunized with mRIPOδ6 or mOVA2 or with poly(I:C) or poly(I:C) + SIINFEKL peptide by i. m. inoculation (as described for Fig. 5a–c). Immunized mice received bilateral tumor implants: ipsilateral B16F10.9-OVA and contralateral B16F10.9 (Fig. 6d). None of the immunization regimens altered the progression of B16F10.9 tumors, lacking OVA (Fig. 6d). Only mOVA2 immunization prevented growth of the ipsilateral B16F10.9-OVA tumors (Fig. 6d, e). Thus, the

**Fig. 5 mOVA2 and mRIPO(H3.3) induced epitope-specific CD8 T cell responses in vivo. a**, **b** IFN-γ ELISPOT showing generation of SIINFEKL-specific CD8 T cells upon i.m. immunization of hCD155-tg mice with mOVA2. Cells were either left unstimulated or stimulated with peptide (SIINFEKL or MOG negative control); Concanavalin A stimulation (positive control) is shown in Supplementary Fig. 14. **c**, **d** H2Kb-SIINFEKL pentamer staining of peripheral blood, gated on CD11b⁻CD19⁻ cells, showing % CD8 T cells that are SIINFEKL-specific. **e**, **f** HLA-A2-RMSAPSTGGV tetramer staining of peripheral blood, gated as in (**c**, **d**), showing % CD8 T cells that are RMSAPSTGGV-specific upon i.m. immunization of AAD_hCD155-tg mice with Complete Freund's Adjuvant (CFA) and RMSAPSTGGV peptide emulsion, or with mRIPO(H3.3). **c**, **f** The gating strategy employed is shown in Supplementary Fig. 15. **a**, **d**, **e** Asterisks denote statistical significance between groups determined by RM-ANOVA protected Tukey's- (a) or 1-way ANOVA protected Sidak's post hoc test (**d**, **e**). All experiments were performed twice and representative results are shown. Error bars denote SEM. **a** mRIPOδ6 ($n = 9$), poly(I:C) ($n = 3$), poly(I:C) + SIINFEKL ($n = 9$), mRIPOδ6 + SIINFEKL ($n = 9$), mOVA2 ($n = 10$). **d** mRIPOδ6 ($n = 9$), poly(I:C) ($n = 3$), poly(I:C) + SIINFEKL ($n = 10$), mRIPOδ6 + SIINFEKL ($n = 10$), mOVA2 ($n = 9$). **e** mRIPOδ6 ($n = 3$), CFA + RMSAPSTGGV ($n = 2$), mRIPO(H3.3) ($n = 3$); $n =$ number of mice. *$P < 0.05$, **$P < 0.01$, ***$P < 0.001$, ****$P < 0.0001$.

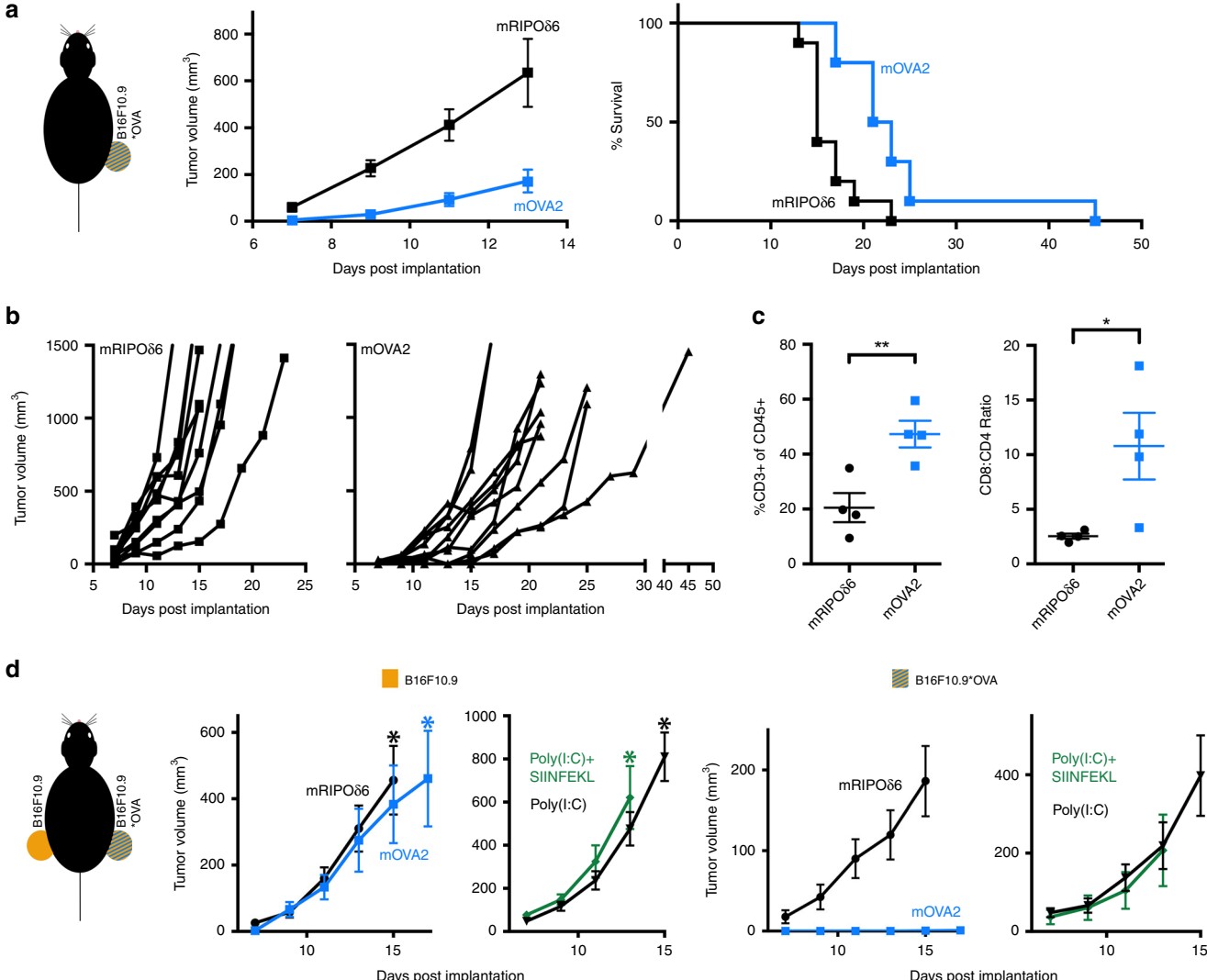

**Fig. 6 mOVA2 immunization delays tumor growth and induces tumor CD8 T cell infiltration. a** Mice were immunized with either mOVA2 or mRIPOδ6 on day 0, boosted on day 14, B16F10.9-OVA cells were implanted subcutaneously 3 weeks later, and tumor volumes were monitored (mice were euthanized when tumors reached 1000 mm³). mOVA2-immunized mice survived significantly longer than mRIPOδ6 treated animals [$P = 0.0017$, Log-rank (Mantel-Cox) test, $n = 10$]. A repeat assay including a mock-immunized group is shown in Supplementary Fig. 18. (**b**) Tumor progression in individual mice from experiment in (**a**). **c** B16F10.9-OVA tumors from mOVA2-immunized mice had elevated CD8 T cell infiltration compared to mRIPOδ6-immunized mice. The gating strategy is shown in Supplementary Fig. 17. Two-tailed t-test, $n = 4$. *$P < 0.05$, **$P < 0.01$. **d** The anti-tumor effect of mOVA2 immunization correlates with SIINFEKL CD8 T cell frequency and is specific to tumors expressing OVA. Mice were immunized with mRIPOδ6, mOVA2, poly(I:C), or poly(I:C) + SIINFEKL as in Fig. 5a, followed by bilateral B16F10.9 and B16F10.9-OVA tumor implantation as shown (left panel). Tumor progression was monitored until reaching >1000 mm³ in the first test animal (asterisks; this occurred in the contralateral B16F10.9 tumors in all treatment groups). mRIPOδ6 $n = 9$, mOVA2 $n = 8$, poly(I:C) + SIINFEKL $n = 8$, poly(I:C) $n = 10$. All error bars represent SEM; $n =$ number of animals.

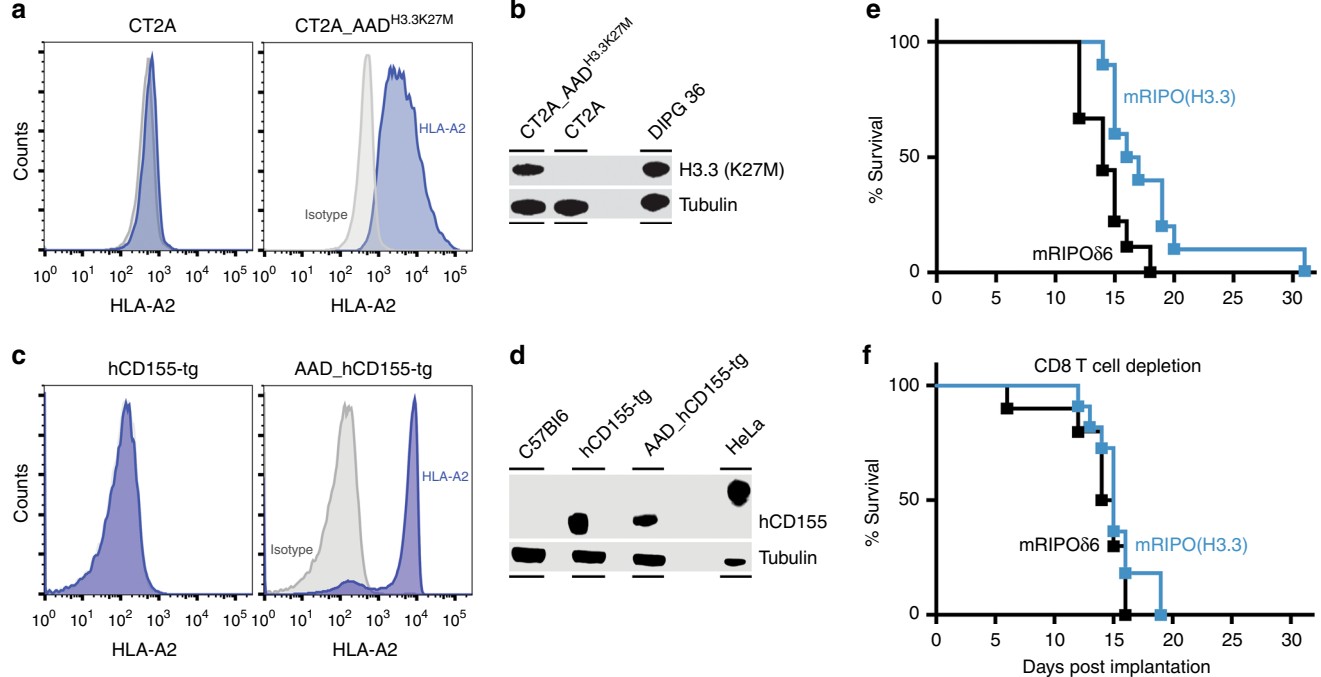

**Fig. 7 mRIPO(H3.3) immunization extends survival in an intracerebral glioma model. a, b** CT2A cells were transduced with HLA-A2 (AAD) (**a**) and full-length mouse histone 3.3(K27M) (b). DIPG 36 is a human H3.3$^{K27M}$ + DIPG cell line used as a positive control. **c, d** AAD_hCD155 transgenic mice express HLA-A2 (AAD) in splenocytes (**c**) and hCD155 (in brain; **d**). HeLa cells were used as a positive control (the differences in hCD155 electrophoretic mobility are due to differential glycosylation). **e** AAD_hCD155 transgenic mice were immunized by i.m. inoculation (day 1), implanted with CT2A_AAD$^{H3.3K27M}$ cells for orthotopic tumor initiation (day 7), boosted with the same regimen (day 14), and followed for assessment of weight and neurological status. Mice were euthanized after losing 15% of their max. weight. mRIPO(H3.3)-immunized mice survived significantly longer than their mRIPOδ6-immunized littermates [$P = 0.0121$, Log-rank (Mantel-Cox) test; mRIPOδ6 $n = 9$ animals, mRIPO(H3.3) $n = 10$]. A replicate is shown in Supplementary Fig. 19. (**f**) CD8 depletion abrogates the anti-tumor effect of mRIPO(H3.3) immunization. The assay was performed as in (**e**), with CD8 depletion antibody administered every 4 days starting at 7 days prior to tumor implantation [$P = 0.3493$, Log-rank (Mantel-Cox) test; mRIPOδ6 + αCD8 $n = 10$ animals; mRIPO(H3.3) + αCD8 $n = 11$].

antitumor effect of mOVA2 immunization is due to its capacity to instigate CD8 T cell responses against a tumor-specific antigen, as evident in ELISPOT and pentamer analyses (Fig. 5a–c).

**mRIPO(H3.3) enhances survival in a mouse glioma model**. We evaluated mRIPO(H3.3) immunization in an immunocompetent malignant glioma model transduced with HLA-A2 (AAD) and the H3.3$^{K27M}$ variant (the gene products of mouse and human *H3F3A* are 100% conserved) (Fig. 7a, b). Of the available syngeneic mouse glioma models, we favor the 20-methylcholanthrene-induced CT2A[41], because it recapitulates the notorious aggressive growth and immunotherapy resistance of the human disease[42]. CT2A_AAD$^{H3.3K27M}$ tumors were orthotopically implanted in AAD_hCD155-tg mice (Fig. 7c, d).

I.m. immunization with mRIPO(H3.3) significantly extended the survival of CT2A_AAD$^{H3.3K27M}$ tumor-bearing mice when compared to mRIPOδ6 immunized mice (Fig. 7e), in a manner similar to mOVA2 immunization against B16F10.9-OVA (Fig. 6a). In a repeat experiment, mRIPOδ6/mRIPO(H3.3)-immunized animals received intraperitoneal inoculations of anti-CD8 antibodies for CD8 T cell depletion (initiated 7 days prior to tumor implantation). This abolished the therapeutic effect of mRIPO(H3.3) immunization (Fig. 7f). Thus, PVSRIPO vector-induced H3.3$^{K27M}$-specific CD8 T cell responses successfully target intracerebral H3.3$^{K27M+}$ malignant gliomas in vivo.

**RIPO(H3.3)-induces type I IFN activation of human DCs**. We evaluated RIPO(H3.3) in primary human DCs to validate the

vector phenotype, and to test presentation of the H3.3$^{K27M}$ antigen to T cells. Infection of human monocyte-derived DCs (from HLA-A2 donors) with RIPO(H3.3) (Fig. 8a; repeat analyses are shown in Supplementary Fig. 20a) did not elicit a cyto-pathogenic program, consistent with the mouse BMDC analyses of mOVA2 (Fig. 3a, b). RIPO(H3.3) translation in human DCs was more efficient than mOVA2 in murine GMCSF-BMDCs, likely reflecting host-specificity (Fig. 8a; compare Fig. 3a). RIPO (H3.3) translation in human DCs closely tracked viral propagation, established in one-step growth curve assays (Fig. 8b). Viral propagation was productive, if limited, with a ~10-fold surge in virus progeny upon initial infection to reach a maximum of ~0.05 pfu per cell by 24hpi (Fig. 8b). Restricted RIPO(H3.3) propagation in human DCs is consistent with lacking cytopathogenicity[9]. Increases in viral progeny were detected up to ~36hpi (Fig. 8b).

RIPO(H3.3)-infected human DCs exhibited IFN-dominant proinflammatory activation and maturation (Fig. 8a, c, d) with sustained type I IFN signatures, e.g. p-STAT1(Y701) and STAT1/ ISG15 induction (Fig. 8a). RIPO(H3.3) infection had similar effects on DC maturation (CD40, −80/86, CCR7) and PD-L1 in human DCs as mOVA2 in mouse BMDCs; however, the effect was more pronounced in PVSRIPO's natural human host (Fig. 8c; compare Fig. 3a, b). RIPO(H3.3) did not affect MHC I/II surface expression in human DCs, as in murine BMDCs (Supplementary Figs. 6a, b; 20b).

Consistent with immunoblot analyses in human DCs (Fig. 8a), and the cytokine response in mOVA2-infected BMDCs (Fig. 3c, d), RIPO(H3.3) infection of human DCs induced a potent type I (IFN-α/ß) and III (IFN-λ1) IFN-dominant response (Fig. 8d;

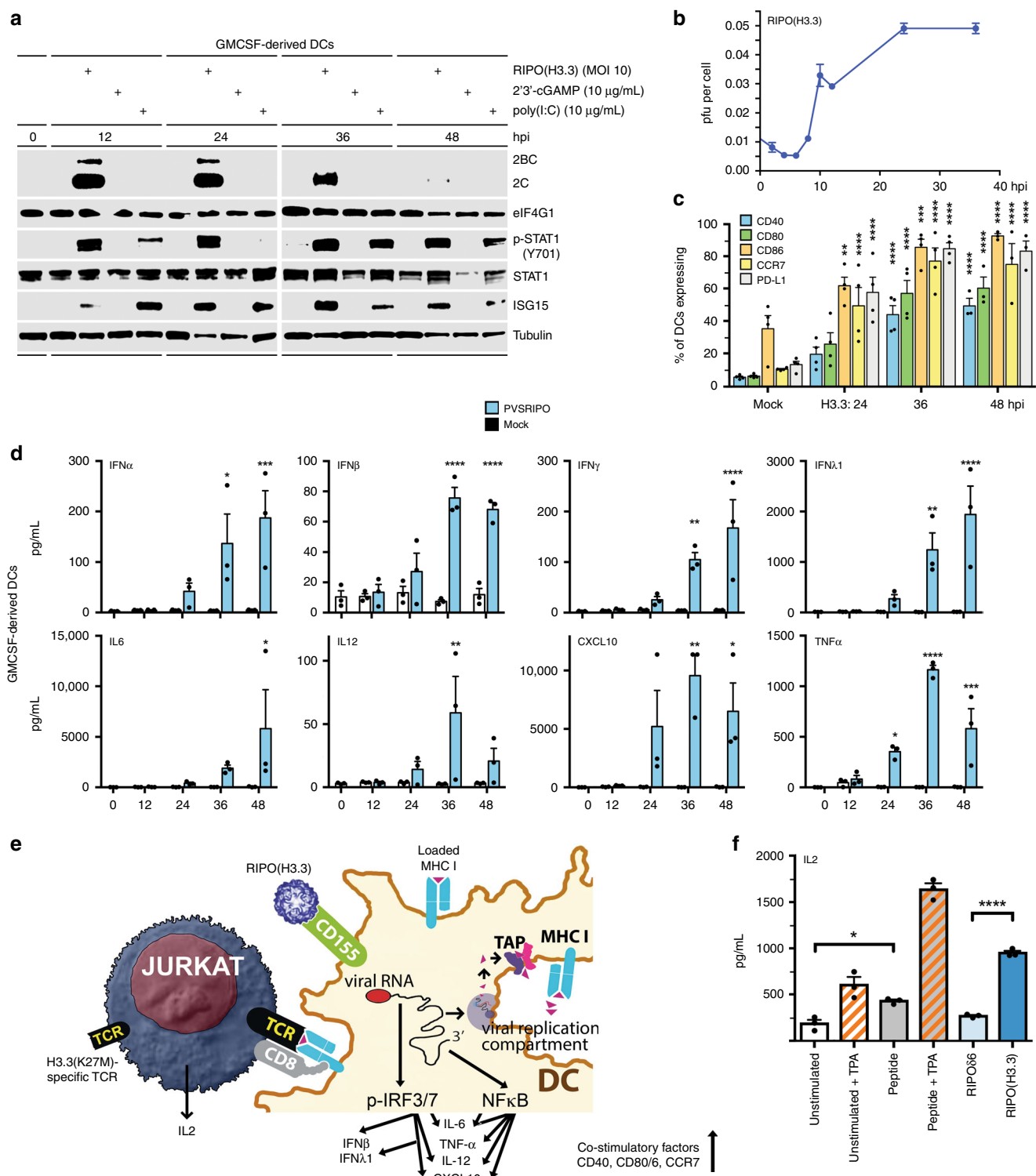

**Fig. 8 RIPO(H3.3) activates human DCs and mediates H3.3^K27M presentation to CD8 T cells. a** Immunoblot of primary human DCs treated with poly(I: C) or cGAMP or infected with RIPO(H3.3). **b** One-step growth curve showing virus replication in primary human DCs ($n = 2$); the experiment was repeated twice and a representative series is shown. **c** Surface co-stimulatory molecule expression on DCs after infection. The gating strategy is shown in Supplementary Fig. 20c. Asterisks depicts significant difference from Mock group as determined by Two-way RM-ANOVA protected Dunnett's post hoc test; 0–36 hpi $n = 4$, 48 hpi $n = 3$. **d** Cytokine induction profile in RIPO(H3.3) infected DCs. Asterisks denote significant difference from Mock 0 group as determined by one-way ANOVA protected Dunnett's post hoc test; $n = 3$. (**e**) Illustration of activation of DCs and antigen presentation to CD8 T cells. **f** Primary DC and Jurkat-TCR co-culture showing RIPO(H3.3)-infected DCs can present the H3.3^K27M antigen to CD8 T cells. Asterisks denotes significant difference between groups as determined by one-way ANOVA protected Tukey's post hoc test, $n = 3$ independent repeats. Experiment was repeated twice. All error bars denote SEM. *$P < 0.05$, **$P < 0.01$, ***$P < 0.001$, ****$P < 0.0001$.

IFN-λ1 was not part of the mouse panel). Compared to mouse BMDCs, RIPO(H3.3)-infected human DCs exhibited a broader proinflammatory response, including IL-6, IL-12, TNF-α, and IFNγ (Fig. 8d). See Supplementary Fig. 21 for complete analyses.

**RIPO(H3.3)-infected DCs present H3.3$^{K27M}$ to CD8 T cells.** Events stemming from RIPO(H3.3)-infection of human DCs indicate broad type I/III IFN-dominant engagement, DC maturation/antigen presentation enhancement, and T cell recruitment and co-stimulation (Fig. 8e). To confirm RIPO(H3.3)-mediated H3.3$^{K27M}$ epitope expression in human DCs and its presentation to CD8 T cells, we used a Jurkat T cell clone expressing CD8 and a H3.3$^{K27M}$-specific TCR[15] (Fig. 8e). Human DCs were left unstimulated, treated with the H3.3$^{K27M}$ peptide, infected with RIPOδ6, or infected with RIPO(H3.3) and co-cultured with the H3.3$^{K27M}$-TCR+, CD8+ Jurkat cells (J76CD8$^+$TCR$^+$) (Fig. 8f). Robust IL-2 release by co-cultured J76CD8$^+$TCR$^+$ only occurred with RIPO(H3.3) infection (Fig. 8f). IL-2 production after DC stimulation with the H3.3$^{K27M}$ peptide was greatly stimulated by adding phorbol ester (TPA) to the culture; in contrast, RIPO(H3.3) infection yielded similar levels of J76CD8$^+$TCR$^+$ activation without TPA (Fig. 8f). Our data summarized in Fig. 8e show that RIPO(H3.3) infects human DCs and provides the 3 required signals for generating antitumor CD8 T cells, i.e. MHC:epitope complexes; maturation marker induction; and proinflammatory cytokines.

## Discussion

Failure of early cancer vaccines—usually short peptides—were attributed to lacking DC engagement[43]. Above all, peptide vaccination without proper DC costimulation and proinflammatory cytokines can induce tolerance or T cell anergy[44–46].

We show that pharmacokinetic problems of peptide vaccination, e.g. poor uptake/presentation by DCs[43], are resolved with PVSRIPO vectors. Due to poliovirus' tropism for CD11c+ antigen-presenting cells[6], PVSRIPO vectors naturally target them for infection. RIPO(H3.3)-vectored epitope expression eliminates the need for specialized antigen cross-presenting cells such as CD141+ (BDCA3+) DCs, required with peptide approaches[47]. Rather, in RIPO(H3.3)-infected DCs, the epitope is loaded onto MHC class I through the classical pathway.

To stimulate DCs in cancer vaccine regimens, many synthetic adjuvants have been enlisted. Yet, these agents incite broad, indiscriminate stimulation, potentially out-of-sync with the kinetics and distribution of epitope presentation. With RIPO(H3.3), type I/III-dominant IFN release and costimulatory activation occur in the very cells expressing the H3.3$^{K27M}$ signature, in step with lingering viral RNA replication. The importance of coincident epitope expression and costimulatory activation in DCs is evident in our studies: CD8 T cell responses to 'empty' mRIPOδ6 (providing the same adjuvancy as mOVA2) combined with SIINFEKL peptide were far inferior to mOVA2.

Type I-dominant IFN responses, elicited by RIPO(H3.3) infection of DCs, provide a fitting costimulatory context for the induction of CD8 T cell responses[48–52]. Protracted, sublethal viral propagation in DCs produced robust, sustained proinflammatory cytokine release, in step with co-stimulatory molecule upregulation and epitope expression. These events are crucial for CD8 T cell priming[53].

PVSRIPO's DC-stimulating phenotype starkly contrasts with the immune evasion and suppression programs of many human pathogenic viruses, which coalesce on DCs and target MHC I functions[54]. Picornaviruses, small RNA viruses with extreme genetic austerity, lack the capacity to interfere with host adaptive immunity. Thus, unlike most human viral pathogens, PVSRIPO does not interfere with MHC I expression, antigen loading or -presentation.

Tumor antigens often resemble self-antigens and are prone to tolerance. The host innate response to picornavirus/PVSRIPO infection is coordinated by the Melanoma-Differentiation Associated Protein 5 (MDA5) pattern recognition receptor[55]. Activation of MDA5 (e.g. by endogenous RNA) has been linked to breaches of self-tolerance and autoimmunity[56]. Thus, MDA5-mediated inflammation may be particularly apt to instigate CD8 T cell responses against weak tumor antigens.

PVSRIPO vectors elicit tumor antigen-specific CTLs capable of infiltrating distant tumors, reduce tumor burden and significantly increase survival in immunocompetent mouse tumor models. The vector 'IRES cassette' accommodates any insert, provided that the salient design principles of our approach are considered. Thus, we outline a clinically feasible enterovirus vector approach based on PVSRIPO that has a clinical track record of safe administration, has empirical evidence for genetic stability, and is capable of unique proinflammatory engagement of DCs for priming of CD8 T cells.

## Methods

**Vector cloning, serial passage, one-step growth curves.** See Supplementary Table 1 for primers. To generate mOVA1, −2 and RIPO(H3.3), the foreign insert was PCR synthesized with primers mOVA1/2-5′ and mOVA1-3′ or RIPO(H3.3)-5′/3′, respectively. A corresponding IRES fragment, spanning from the *Pml*I site in the cloverleaf of mRIPO (mRIPOδ6, mOVA1/2)[9] or PVSRIPO [RIPO(H3.3)] to the conserved cryptic AUG was generated with primers IRES-5′/3′. The mOVA2-flag insert was generated as described above, with the mOVA2-flag-5′ primer instead of mOVA2-5′. mOVA2s carrying an upstream AUG in poor Kozak context, or a frameshift in the foreign insert, were generated as described above with primers mOVA2-PC-5′ or mOVA2-FS-5′ substituting for mOVA2-5′, respectively, and primer mOVA2-PC-3′ replacing IRES-3′ for the former. Fusion PCR with the foreign insert + IRES fragment as templates using flanking primers IRES-5′/mOVA1-3′ (mOVA1/2 and derivates) or IRES-5′/RIPO(H3.3)-3′ [RIPO(H3.3)] yielded segments encompassing the IRESs/foreign inserts for mOVA1/2 and RIPO(H3.3). mRIPOδ6 was generated using IRES-5′ and IRESδ6-3′ and the mRIPO IRES as template. These were inserted into the full-length cDNA clone of PVSRIPO digested with *Pml*I-*Sac*II (mRIPOδ6, mOVA1/2 and derivates) or *Eco*RI-*Sac*I [RIPOH3.3]. In vitro synthesis of infectious RNA was performed using MEGAscript T7 (Thermo, AM1334) for deriving vectors[23,24]. For serial passage, $3 \times 10^6$ HeLa cells in 60 mm dishes were infected with vector (MOI 1; 24 h), freeze-thawed, the resulting lysate centrifuged to pellet debris, and 100 µl of the supernatant used to infect another HeLa cell culture (MOI of ~1). This procedure was repeated 20 times. Total RNA from infected cells at passages 3, 10, and 20 was isolated using TRIzol (ThermoFisher), reverse transcription was performed using Superscript IV (ThermoFisher) with primer Seq-3′, and the region encompassing the entire IRES, sequences coding for the foreign epitope and the adjoining viral polyprotein were PCR-amplified from cDNA with primers IRES-5′ and Seq-3′ for sequencing. One-step growth curves were performed according to established standards[23,24].

**Rodent tumor models, immunization, T cell depletion.** All procedures involving vertebrate animals were performed under a Duke IACUC-approved protocol. Homozygous hCD155-tg C57Bl6 mice are maintained as a breeding colony; they were originally obtained from S. Koike[57]. $5 \times 10^5$ B16F10.9-OVA murine melanoma cells were implanted orthotopically in hCD155-tg mice[9]. AAD-tg C57Bl6 mice (Jackson Labs, strain #004191) express an interspecies hybrid class I MHC gene, AAD, which contains the α-1 and -2 domains of the human HLA-A2.1 gene and the α-3 transmembrane and cytoplasmic domains of the mouse H-2Db gene, under the direction of the human HLA-A2.1 promoter. Heterozygous hCD155- and AAD(HLA-A2)-tg C57Bl6 mice were obtained by cross breeding hCD155-tg mice with AAD-tg mice. Murine CT2A$^{AAD\_H3.3K27M}$ malignant gliomas were implanted intracerebrally in hCD155/AAD-tg mice as follows. Mice were anesthetized (isoflurane), scalp sterilized (betadine and 70% ethanol) and an incision was made along the sagittal suture to expose the bregma. A hole was drilled in the skull near the right coronal suture (2 mm right/0.5 mm anterior to the bregma) and a 30 G needle was used to inoculate $10^5$ CT2A$^{AAD\_H3.3K27M}$ cells in 5 µL methylcellulose suspension (at 3.33 µL/min) at a depth of 3.6 mm. Mice were monitored for clinical symptoms and neurological deficits, and weighed daily after tumor implant; mice were euthanized when they had lost 15% of their max. weight or when they became symptomatic as defined in our IACUC-approved protocol. For i.m. vector immunizations, mice 6–12 weeks of age were inoculated in the gastrocnemius muscle ($10^4$–$10^6$ pfu). A boost dose was administered on day 14 in

the contralateral leg ($10^5$–$10^7$ pfu). For SIINFEKL immunization, 50 µg of poly(I:C) (high molecular weight VacciGrade; Invivogen) was mixed with 50 µg of SIINFEKL peptide (Invivogen) for immunization/boost as described above. For H3.3$^{K27M}$ peptide immunization, 50 µg of RMSAPSTGGV (Peptide 2.0) was emulsified in Complete Freund's Adjuvant (Sigma) and administered intraperitoneally. A boost dose was administered on day 14 with 50 µg of RMSAPSTGGV emulsified in Incomplete Freund's Adjuvant (Sigma). For CD8 T cell depletion, anti-CD8 antibody (BioXCell, BE0061) was administered i.p. (250 µg/dose), starting 7 days before tumor implantation, and continuing every 4 days thereafter.

**Immunoblot/immunoprecipitation, flow cytometry, and IHC.** Immunoblots and immunoprecipitation were performed using standardized protocols with the antibodies listed in Supplementary Table 2[58]. For flow cytometry, cells were washed with- and resuspended in 100 µL FACs buffer (phosphate-buffered saline, 2% fetal bovine serum). The cells were Fc-blocked (TruStain, Biolegend), stained (1 h) with the appropriate antibodies (Supplementary Table 2), washed and resuspended in 250 µL FACs buffer to be analyzed on a BD LSRFortessaX20. Compensation and data were analyzed using FlowJo. H2Kb-SIINFEKL pentamer (Proimmune) staining was performed with blood seven days post boost-immunization according to manufacturer's protocol. The HLA-A2-RMSAPSTGGV tetramer was synthesized by the NIH Tetramer Core Facility. hCD155_AAD-tg mice were immunized with vector or peptide and bled one week after boost. Blood was lysed with ACK Lysing buffer (Gibco, A1049201), washed (FACs buffer), stained with tetramer (30 min, 20 °C), washed, Fc-blocked (see above), stained with CD8, CD19 and CD11b antibodies and analyzed as described above. For IHC, lymph nodes were harvested from euthanized mice, fixed in 4% parafomaldehyde, dehydrated in 70% ethanol, paraffin-embedded and sectioned (7 µM). Mounted sections were stained using the Ventana Discovery Ultra platform (Research Immunohistology Lab, Duke Dept. of Pathology).

**ELISA, cytokine bead array, and ELISPOT.** We used Ready-Set-Go ELISA kits (ThermoFisher) for Granzyme B, IFNγ and (human) IL2 ELISAs. Cytokine Bead Arrays were performed using Biolegend mouse (#740621) and human (#740390) antivirus LEGENDplex kits. Mouse skeletal muscle tissues were minced in 500 µL of DMEM medium, centrifuged, and the resulting supernatants were used for LEGENDplex analyses. For ELISPOTs, plates (Millipore, MSIPS4W10) were pre-activated with 35% ethanol, washed 6 times with PBS, and coated with 10 µg/mL of antibody [IFNγ; AN-18 (Mabtech)] (12 h, 4 °C). Plates were blocked with R10 medium [RPMI, 10% FBS, non-essential amino acids, 1 × 2-mercaptoethanol (Gibco 21985023)] (2 h, 37 °C) prior to splenocyte addition. Spleens were processed into single cell suspension, red blood cells lysed using ACK Lysing buffer, and splenocytes washed with R10 medium. Cells were plated at a density of $5 × 10^5$/well in 100 µL of R10 in triplicate, and remained unstimulated or stimulated with 10 µg/mL of peptide [OVA$^{257-264}$ (Invivogen); myelin-oligodendrocyte glycoprotein (MOG)$^{35-55}$ (Sigma)]; 4 µg/mL Concanavalin A (Sigma) and incubated (24 h, 37 °C). After incubation, the plates were washed with PBS/0.05% Tween, incubated with biotin-labeled anti-mouse IFNγ [R4-6A2 (Mabtech); 1 µg/mL] (2 h, 37 °C), washed, incubated with Avidin Peroxidase Complex (VectaStain, 1 h) and developed with AEC ELISpot substrate. All ELISPOT assays were evaluated externally by Zellnet Consulting.

**Cell lines, primary cultures, co-culture assays.** *Cell lines:* We used HeLa R19- and HEK293 cells for virus propagation and one-step growth curve assays[28]. B16F10.9 murine melanoma cells were obtained from ATCC; derivation of B16F10.9-OVA was described elsewhere[9]. CT2A cells were kindly provided by Dr. P. Fecci (Duke Univ.); the CT2A stock was validated by whole exome genome sequencing. CT2A_AAD$^{H3.3K27M}$ cells were derived by transfecting CT2A cells with linearized AAD (Addgene #14906)[59] cDNA, followed by transduction with lentivirus expressing H3.3$^{K27M}$ (a gift from Dr. H. Yan, Duke Univ.). CT2A_AAD$^{H3.3K27M}$ cells were sorted to select for HLA-A2+ cells and H3.3$^{K27M}$-expressing cells were selected with hygromycin (2 weeks at 100µg/mL). DIPG 36 cells were generously provided by Dr. M. Monje (Stanford Univ.). The Jurkat T cell line (J76CD8 + TCR+) was generated by lentiviral transfection of J76CD8+ cells[60] with the cDNA of a TCR with high affinity for the H3.3$^{K27M}$ epitope (RMSAPSTGGV) isolated from PBMCs of an HLA-A2+, H3.3(K27M)-mutated DIPG patient[15]. *Primary cultures:* Mouse bone marrow-derived dendritic cells (BMDCs) were generated from bone marrow cells extracted from femurs and tibias dissected from hCD155-tg C57Bl6 mice. Bones were flushed out bone marrow, red blood cell lysed with ACK Lysing buffer and cells were washed with R10 medium. For GMCSF-BMDCs: cells were counted and plated at $10^6$ cells/mL, supplemented with IL-4 (10 ng/mL; Sigma, I1020) and GMCSF (20 ng/mL; Sigma, G0282). On day 3, fresh R10 medium with IL-4/GMCSF was added. On day 7, the loosely adherent cells were harvested and re-plated at $10^6$ cells/mL for subsequent experiments. For FLT3L-BMDCs: cells were plated at $2.5^6$ cells/mL in R10 medium supplemented with 300 ng/mL FLT3L (ThermoFisher, PHC9415) for 9 days[33]. All BMDC preparations were tested for CD11c expression by flow cytometry. Human monocyte-derived DCs (human DCs) were derived from PBMCs obtained from Stem Cell Technologies (#70025) briefly, monocytes were cultured with

GMCSF/IL4 for 6 days in AIMV medium[9,61]. *OT-I CD8 T cell/BMDC co-culture:* $10^5$ OT-I CD8 T cells (isolated from OT-I transgenic mouse spleen (Jackson Laboratories #003831) using the Biolegend CD8 T cell isolation kit #480008) and $10^5$ GMCSF-BMDCs (with appropriate treatment) were cocultured for 3 days in a 96-well U-bottom plate. Supernatant was harvested and tested for Granzyme-B and IFN-γ by ELISA. *J76CD8 + TCR + T cell/hDC co-culture:* J76CD8 + TCR+ cells were sorted using CD8 and Tetramer+ (BD DiVa Sorter, Duke Cancer Institute Flow Cytometry Core) and co-cultured with hDCs (HLA-A2+) treated with either virus, peptide or mock were cocultured with J76 cells (48 h) in U-bottom 96 well plate. In co-culture experiments with peptide stimulation, 12-O-tetra-decanoylphorbol-13-acetate (TPA; Sigma Aldrich) was added (10 ng/mL). IL-2 ELISA was performed with supernatant after 48 h.

**Statistical analysis.** Statistical analyses were preformed using GraphPad Prism. Error bars represent Standard Error of the Mean (SEM). Significance was determine dusing two-tailed Student's $t$ test when comparing 2 groups; for multiple groups, an ANalysis Of VAriance (ANOVA) was first performed followed by the appropriate post hoc test as described in the figure legends. The number of independent replicates (n) and number of times experiments were repeated are indicated in the figure legends.

**Reporting summary.** Further information on research design is available in the Nature Research Reporting Summary linked to this article.

## Data availability

All data supporting our findings are included within the Manuscript and Supplement. The source data underlying all figures in this manuscript are provided as Source Data files.

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

## Acknowledgements

The authors thank Malte Mohme (University of Hamburg Medical School, Germany), Adam Swartz and David Boczkowski (Duke University) for technical advice and discussion. DIPG 36 cells were generously provided by Michelle Monje (Stanford University). This work was supported by PHS Grants R01 NS108773 (M.G., S.K.N.) and F32 CA224593 (M.C.B.), a Defeat DIPG Research Grant (D.A.), and a Research Grant from the V Foundation (D.A., M.G.).

## Author contributions

M.M.M., M.G., M.C.B., S.K.N., D.A., and D.D.B. contributed to the conception and design of the study. M.M.M., E.Y.D., Y.Y. and J.C. performed experiments and carried out statistical analyses. M.M.M., E.Y.D., M.C.B., and M.G. drafted the original manuscript which was reviewed and revised by all authors. H.O. contributed critical sophisticated reagents, assays, and technical advice.

## Competing interests

E.Y.D., M.C.B., S.K.N., D.D.B., D.A., and M.G. are co-inventors of intellectual property that was licensed to Istari Oncology, or may be licensed in the future. M.G. and D.D.B. are co-founders of-, advisors to-, and equity holders in Istari Oncology.
