## [Peer Review File · Nature Communications]

Reviewers' comments:

Reviewer #1, expert in clinical oncolytic virus therapies (Remarks to the Author):

The authors describe the generation of a Poliovirus expressing the H3.3 antigen or antigen-marker OVA, shown to be genetically stable in vitro and able to present the H3.3 or OVA epitope in DCs after infection. Intramuscular injection of the OVA-expressing poliovirus in mice resulted in increased immune infiltration and activation and rejection of a murine tumor overexpressing (and presenting) the OVA-antigen. The authors also did elaborate in vitro studies to show that DCs infected with the antigen-expressing poliovirus vector lead to an activation pattern, similar to that by caused LPS induction. Although this manuscript and number of experiments are extensive, the antigen-delivery strategy by poliovirus as a cancer immunotherapy is not elucidated/addressed. Other oncolytic viruses have been used to express tumor antigens and thus the only novelty here is the use of poliovirus instead of another virus. Further, the authors utilize a strong surrogate antigen (OVA) and test non-DIPG models for most of their experiments, although the introduction, title and abstract hypothesize that they want to test the H3.3 antigen for DIPGs, but there is little data on this.

Concerns:

- 1) The authors show that they are able to generate a genetically stable Poliovirus Vector that expresses a transgene/antigen. The authors however did not evaluate transgene expression in vivo or show if it is expressed.
- 2) The authors indicate that DCs are be the main target of the Poliovirus vector, but there is no data on in vivo infection, vector replication data or antigen expression. Other factors and immune population contribute to antigen-presentation towards T cells. For example, NK cells should be able to purge virally infected DCs and DCs still need to efficiently home to the lymph nodes. There is a lot of extrapolation of in vitro data assuming that it will also be valid in an in vivo setting.
- 3) The authors do provide evidence that inducing a T cell response against OVA can repress a melanoma cell line expressing the OVA-antigen. Albeit the question remain whether this approach can increase rejection of a less immunogenic cancer, such as GBM or DIPG. Further, GBM patients are known to have a more exhausted T cell response and thus might be much less responsive towards the treatment. Whether a single-antigen expressed by the poliovirus in a GBM or DIPG setting, as currently suggested, is able to be of therapeutic value is questionable.
- 4) Vaccination strategies often include prime/boost administration. Adaptive immunity against the poliovirus in the prime pahse might decrease its efficacy as an antigen-deliver vector.
- 5) Most of the validation is carried out the OVA antigen, a very strong foreign antigen. It remains debatable whether responses against weaker tumor antigens, such as DIPGs, will be as effective.
- 6) Figure 5: ELISPOT and IFN-gamma staining lack a positive control, providing additional information that the potential to secrete IFN-gamma is maintained and percentage of IFN-gamma producing cells is similar over all samples.
- 7) Figure 6: An Untreated control is lacking from the figure. This tumor rejection model is modelling melanoma and does not provide information concerning a more immunogenic cold tumor, such as GBM or DIPG, nor does it provide information whether this immune response will be limited by the blood-brain barrier.
- 8) The evidence connecting T cell priming and polio is not that strong. Polio does not stimulate (or mature) antigen presenting DC as IL12 is negative (in Fig S3 and Fig 3) while the IFN pathway appears to be stimulated. This implies that pDC, cDC or macrophages are most likely the target of Polio. I also question the FACS analysis, where the gating method seems inconsistent through experiments with even some errors (7AAC positive = dead, but same figure S4 says live cells).
- 9) I question the vaccine strategy of Fig5. IM immunization requires draining Lymph Nodes for proper T cell population stimulation. OVA-T cells in splenocyte and blood suggests mOVA2 viruses are leaked systemically as T cell activation in the control polyIC:SIINFEKL was negligible, but should be more positively activated.
- 10) The response of Figure 6B is not durable and not that impressive, again questioning vaccine efficacy.

Reviewer #2, expert in oncolytic virus design (Remarks to the Author):

These authors have published extensively on the oncolytic virus poliovirus:rhinovirus chimera PVSRIPO. This virus has demonstrated promise in clinical and preclinical studies. In this manuscript, the authors report on modification the poliovirus:rhinovirus chimera PVSRIPO in the stem loop region to promote the stable expression of exogenous antigens, while preserving the pre-existing attributes for this vector including the following:

1. tropism for antigen presenting cells
2. no interference with immune activation (unlike many viruses with immune avoidance genes)
3. antigen presentation
4. robust engagement of inflammatory responses

The authors convincingly demonstrate that these stem loop modifications did not adversely affect the intended translation initiation, the foreign transgene insert was expressed and the product was processed properly.

In addition, the authors designed and implemented various in vitro and in vivo experiments to demonstrate that type I IFN was induced by the engineered virus, antigen presentation occurred, and that tumor antigen-specific CD8 T cells were induced and activated by treatment with the antigen-expressing virus. Finally, the authors demonstrate that CD8 T cells migrated into the tumor site, and that this infiltration was associated with a reduction in tumor burden and an increase in animal survival in one mouse model.

Generalizability: Given the limited scope of tumor models explored, and limited foreign antigen transgene constructs evaluated, the likely generalizability of this approach to other systems and antigens cannot be assessed.

Technical questions & comments:

While the authors have shown that there was genetic stability preserved in the foreign insert after 20 passages of virus in cells in culture, they did not comment on the stability in the rest of the viral genome. Did they sequence the whole viral genome to rule out any potential change in other regions of the genome that were not around the locus of the foreign insert? This could have significant safety implications.

The authors should also consider evaluating this system in additional tumor models, and with additional antigen transgene constructs to demonstrate the likely generalizability of this approach to other systems and antigens.

In addition, it is likely that the oncolytic effect of this virus will be reduced by increasing its immunogenicity. This question should be addressed by evaluating the relative efficacy of a tumor antigen expressing PVSRIPO in a tumor model demonstrating potent oncolytic virus replication and efficacy.

Reviewer #3, expert in clinical glioma immunotherapy (Remarks to the Author):

In this manuscript, Mubeen et al. designed a poliovirus/rhinovirus chimera-based vector, named mOVA2, which infects dendritic cells(DCs) such that antigen peptide (epitope) carried by the virus can be expressed by DCs, followed by antigen presentation to cytotoxic CD8+ T cells. mOVA2 overcomes the neuropathogenicity and genetic instability originated from the poliovirus, while maintaining its

tropism to DCs.

The authors demonstrated that mOVA2 infects, activates and induces epitope presentation in mouse bone marrow derived dendritic cells (BMDCs) and the following activation of CD8+ T cells in vitro. In addition, mOVA2 recruits and activates DCs, and triggers antigen-specific T cell activation in vivo. Using a murine model of melanoma, they showed that mOVA2 recruits CD8+ T cells to the tumor, reduces tumor size and increases survival.

As a proof of concept experiment, Mubeen et al. engineered the virus [named RIPO(H3.3)] to express epitope containing H3.3-H27M, a signature mutation found in patients with diffuse intrinsic pontine glioma (DIPG). The authors showed that RIPO(H3.3) activates human BMDCs and triggers responses from H3.3-K27M-TCR+, CD8+ Jurkat cells. RIPO(H3.3) has a good translational value given that intratumoral infusion of PVSRIPO virus in patients with recurrent grade IV malignant glioma confirmed absence of neurovirulence and increased survival in past studies.

The study is of great potential clinical and scientific significance. However, the writing is so filled with jargon and lacking in crucial details that it is very difficult to follow the manuscript.

The main concern is that the authors need to experimentally validate that this viro-immunotherapy shows preclinical efficacy in a valid experimental model of DIPG in vivo. I am unclear what the B16F10.9-OVA tumor model really is, but I do not believe it is a DIPG model.

And the manuscript needs to be edited substantially; I refer the authors to the following guide on scientific writing:

http://www.georgegopen.com/uploads/1/0/9/0/109073507/gopen___swan_sci_of_sci_writing_am_sci_1990_.pdf.

Please see the following specific suggestions:

1. Fig. 2C: a lysate immunoblot showing the total Flag-tagged proteins should be provided for the IP experiment. In addition, what are the reasons for disappearance of the three flag-tagged bands at 6 and 8 hpi?
2. Fig. 3E: what are the results when the same experiments are done in FLT3L BMDCs?
3. In addition to eIF4G1 cleavage, are proliferation, cell death, and senescence affected in the infected BMDCs?
4. There is no reference or introduction for the B16F10.9-OVA melanoma tumor model. How is this relevant to H3.3K27M+ glioma?
5. Fig. 7F: what are the levels of Granzyme B and IFN-gamma in this co-culture system?
6. Is it possible to set up a co-culture system of human BMDC, CD8+ T cells and HLA-A2+ DIPG cells to demonstrate the efficacy of RIPO(H3.3)?
7. A T cell migration assay should be performed with conditioned medium collected from RIPO(H3.3)-infected DCs to demonstrate effective T cell recruitment.
8. It would be nice if the virology terms (e.g. the processing of VP0, VP3 and VP1 and P2/2BC/2C) are explained for readers not in this field.
9. Sup Fig.1: please show experimental data demonstrating the viability of each vector.

Minor comments:

The statement "Unlike adult GBM, which is extremely heterogeneous, ~80% of Diffuse Intrinsic Pontine Gliomas (DIPG) and ~20% of pediatric GBMs carry a homogeneously expressed driver mutation in histone 3.3 [H3.3K27M]12, 13" is neither complete nor entirely accurate. While it is true that 80% of DIPGs express H3K27M mutation, hemispheric pediatric GBMs express a different histone mutation in ~20% cases, H3G34R/V. Furthermore, other midline gliomas such as thalamic and spinal cord, also express the H3K27M mutation. This entity is now classified as H3K27M+ diffuse midline glioma (DMG) by the WHO and this formal reclassification would be referenced.

Reviewer #4, expert in preclinical development of oncolytic virus (Remarks to the Author):

This is a clearly written and presented manuscript that argues that a picornavirus vector created by the authors can be used as a T cell vaccine specifically for use in cancer indications. The authors argue that while in general picornavirus vectors are not useful for expressing transgenes due to inherent instability, their design of incorporating a peptide sequence into a critical regulatory region of the virus will favour retention of transgenic sequences. This is a strategy that this same group published on in 2003 (Dobrikova et al). They present very nice data that shows their clinically validated vector can stimulate respectable T cell responses against encoded peptides.

Critique

- (1) the authors do not really provide any detailed discussion comparing and contrasting the results from this paper and their 2003 work. Is this an incremental improvement compared to what they have done previously or a significant improvement and why?
- (2) The observation that the SINNFEEKL peptide encoding sequence evolved over time suggests that there may be significant restrictions on what can be encoded...this should be discussed.
- (3) the virus cannot encode more than one peptide? Is this the case? These types of limitations of the technology should be discussed.
- (4) The authors present data that shows about 0.5% of T cells recognize the encoded peptide after PVRIPO immunization. Can this be boosted with more administrations. Do the authors feel this is a clinically relevant level of T cell stimulation and if so why?
- (5) Most of the population has already been vaccinated against polio, will this impact the ability of this vector to initiate immune responses?

Reviewer #1, expert in clinical oncolytic virus therapies (Remarks to the Author):

The authors describe the generation of a Poliovirus expressing the H3.3 antigen or antigen-marker OVA, shown to be genetically stable in vitro and able to present the H3.3 or OVA epitope in DCs after infection. Intramuscular injection of the OVA-expressing poliovirus in mice resulted in increased immune infiltration and activation and rejection of a murine tumor overexpressing (and presenting) the OVA-antigen. The authors also did elaborate in vitro studies to show that DCs infected with the antigen-expressing poliovirus vector lead to an activation pattern, similar to that by caused LPS induction. Although this manuscript and number of experiments are extensive, the antigen-delivery strategy by poliovirus as a cancer immunotherapy is not elucidated/addressed. Other oncolytic viruses have been used to express tumor antigens and thus the only novelty here is the use of poliovirus instead of another virus. Further, the authors utilize a strong surrogate antigen (OVA) and test non-DIPG models for most of their experiments, although the introduction, title and abstract hypothesize that they want to test the H3.3 antigen for DIPGs, but there is little data on this.

The concept of viral vectors is >40-years old and we acknowledge that numerous strategies have been developed to deliver antigenic material with replication-competent viral constructs ('vectors') before.

However, we contend that the strategy described in this research is fundamentally distinct from that of previously investigated vectors for the following reasons:

1. Clinical application: the backbone for our vector construct, PVSRIPO, has been tested with intracerebral inoculation (the most stringent route of administration for evaluating poliovirus neuro-attenuation) in phase I testing in human subjects, where it was found to have a favorable safety profile.¹¹ Based upon the results herein we are planning an IND submission, enabling clinical trial to test our design in (H3.3^{K27M}-positive) DIPG patients.
2. Unique tropism of poliovirus and PVSRIPO that prominently includes antigen presenting cells (macrophages and dendritic cells).¹²
3. An unusual sub-lethal vector phenotype in host antigen presenting cells, producing lingering, non-cytotoxic infection and a powerful type I/III interferon-dominant pro-inflammatory signature.

4. Lack of interference with antigen presentation/lack of immune subversive /immunosuppressive mechanisms.
5. Targeting of the DIPG antigen H3.3^{K27M}.
6. Successful design of a *genetically stable* enterovirus vaccine vector prototype.

Many proposed vaccine vector strategies have enlisted DNA viruses, as their large coding capacity and relative genetic stability allows for the incorporation of substantial foreign inserts. Our own strategy has a disadvantage in this regard, as our vector platform is restricted by limited coding capacity.

However, dsDNA viruses commonly used as vector backbones, e.g. alpha-herpesviruses, adenoviruses or poxviruses, are notorious for suppressing antigen-presentation. A myriad of mechanisms of how they accomplish this have been described, e.g. blocking MHC expression and subcellular trafficking, preventing antigen loading onto MHC (e.g. by interfering with TAP1 function), blocking antigen presentation on the cell surface, suppressing dendritic cell migration or outright dendritic cell killing. These phenomena are extensively reported and reviewed in the literature [see for example Petersen *et al.* (2003) Virus evasion of MHC class I molecular presentation. *J Immunol* 171:4473].

This is an enormous problem with viral vectors, as antigen presenting cells are essential for instigating antigen-specific adaptive immunity.

This conundrum forms the base for our premise. What we describe in much detail in our studies *in vitro* and *in vivo*, is a highly unusual host:virus relationship between dendritic cells and PVSRIPO-based vectors. The sublethal PVSRIPO phenotype in infected dendritic cells: (i) prevents viral damage to dendritic cells that could interfere with their antigen-presenting and T cell co-stimulating functions (see detailed response to **comment #3** by **Reviewer 3**, pg. 23-24); (ii) allows for (vector-driven) expression of the H3.3^{K27M} epitope *within* dendritic cells (see our detailed responses to **comments #1, #2** below); (iii) elicits a powerful, sustained type I, type III IFN-dominant proinflammatory response *within* dendritic cells. There is overwhelming empirical support from countless studies that intrinsic type I IFN responses are essential drivers of dendritic cell function in adaptive immunity, e.g. in the context of antitumor immunity.¹³

Thus, rather than suppressing or diverting adaptive immunity by interfering with antigen presenting cells (a domain of DNA viruses), PVSRIPO-based vectors *actively stimulate* antigen presentation and T cell co-stimulation. The reasons for this are thoroughly documented in our study; they relate to enterovirus' lack of immune-evasive/suppressive capacity and, perhaps, the specific features of PVSRIPO imparted by the foreign rhinovirus IRES.

The final conclusion of our study is that PVSRIPO-based vectors are capable of achieving the coveted triad of *antigen expression, type I/III IFN-dominant pro-inflammatory stimulation, and maturation marker induction in the same (infected) dendritic cell* (see Discussion). We have accumulated compelling empirical evidence for this scenario *in vitro* and *in vivo*.

This is a significant result which may deliver fundamental advances towards better immunization strategies with enhanced therapeutic or prophylactic efficacy (e.g. for immunizing against the H3.3^{K27M} signature of DIPG).

Concerns:

“1) The authors show that they are able to generate a genetically stable Poliovirus Vector that expresses a transgene/antigen. The authors however did not evaluate transgene expression in vivo or show if it is expressed.”

and

“2) The authors indicate that DCs are be the main target of the Poliovirus vector, but there is no data on in vivo infection, vector replication data or antigen expression. Other factors and immune population contribute to antigen-presentation towards T cells. For example, NK cells should be able to purge virally infected DCs and DCs still need to efficiently home to the lymph nodes. There is a lot of extrapolation of in vitro data assuming that it will also be valid in an in vivo setting.”

The reviewer raises valid points. To address these issues, we have now included results of immunohistochemistry assays of the H3.3^{K27M} epitope in popliteal lymph nodes, draining the mRIPO(H3.3) immunization site of hCD155-tg mice (see **new Figure 4d**). Our results show that mRIPO(H3.3)-infected cells expressing the H3.3^{K27M} epitope home to the lymph nodes that drain the mRIPO(H3.3) immunization site in skeletal muscle.

In combination with the observed robust antigen-specific T cell priming following vaccination with our OVA vector of hCD155-tg mice (which requires expression of the antigen *in vivo*) these findings have important repercussions for several reasons:

- (i) they validate our *in vitro* data with PVSRIPO-based vectors in mouse GMCSF/FLT3-differentiated BMDCs (**Figure 3**) and in human dendritic cells (**Figure 7**);
- (ii) they confirm our hypothesis that mRIPO(H3.3) targets dendritic cells for infection *in vivo* (as polioviruses do naturally);
- (iii) they show that expression of the H3.3^{K27M} epitope occurs in the intended target, dendritic cells, after intramuscular immunization *in vivo*;
- (iv) they extend our findings of dendritic cell recruitment (**Figure 4b**) and proinflammatory stimulation (**Figure 4c**) after intramuscular immunization with PVSRIPO-based vectors *in vivo*;
- (v) they indicate that [mRIPO(H3.3)-infected] dendritic cells migrate to draining lymph nodes *in vivo*, corroborating our data with mRIPO(H3.3) infection of human GMCSF-differentiated dendritic cells, e.g. upregulation of the lymph node-draining chemokine (CCR7) (**Figure 7c**).
- (vi) they show that antigen expression occurs in immunization site-draining lymph nodes after immunization *in vivo* with mRIPO(H3.3). Antigen expression in this locale is crucial for antigen presentation and the generation of antigen-specific CD8 T cells; corroborating our data on CD8+ T cell responses to antigens delivered by PVSRIPO-based vectors, evident in ELISPOT, pentamer and tetramer assays (e.g. tetramer staining for the H3.3^{K27M} epitope in blood of immunized mice; **new Figure 5e, f**).

Our findings resonate with an important recent report by Shen *et al.*, demonstrating that CD11c+ myeloid cells (macrophages/dendritic cells) are a high priority compartment for poliovirus infection/propagation after oral infection in a representative non-human primate model.⁷ We now show that CD11c+ myeloid cells are also targeted by mRIPO(H3.3) for infection and exhibit expression of the H3.3^{K27M} epitope (**new Figure 5e, f**) and induction of maturation markers (**Figure 4c**) *in vivo*.

Polioviruses, exclusively human pathogens with razor-sharp cell type-specificity (mediated by the human poliovirus receptor CD155), target a limited spectrum of host cells for infection and viral replication. Shen *et al.* showed that, in susceptible non-human primates (*Cynomolgus*, a WHO-validated representative primate model for polio), enteric epithelial cells and CD11c+ macrophages/dendritic cells sustain productive poliovirus replication after oral infection (the natural route).⁷ Given that macrophages/dendritic cells are one of very few cellular compartments permitting natural poliovirus infection and replication in its natural host, it is compelling to assume that CD8 T cell responses against PVSRIPO vector-delivered antigens involve infection of this compartment. Our new data reporting H3.3^{K27M} epitope expression in cells homing to lymph nodes draining the mRIPO(H3.3) immunization site (**new Figure 5e, f**) corroborate this hypothesis.

We agree with the reviewer, in principle, that other antigen presenting cells could present the H3.3^{K27M} epitope to T cells in mRIPO(H3.3)-immunized hCD155-tg mice.

To directly document antigen-specific CD8 T cell responses elicited by PVSRIPO vector-infected dendritic cells, we performed a ‘dendritic cell vaccine’ assay, with (*ex vivo*) mOVA2-infection of FLT3L-differentiated BMDCs adoptively transferred into naïve mice (**new Supplementary Figure 15**; see text in **revised Results**, pg. 13). This produced SIINFEKL-specific CD8 T cell responses, showing that infection of dendritic cells with PVSRIPO-based vectors *in vivo* is sufficient to generate (vector encoded) antigen-specific CD8 T cell responses. It supports our premise that PVSRIPO vector infection of the natural poliovirus target, CD11c+ dendritic cells,⁷ is central to instigating adaptive antigen-specific immunity *in vivo*. In this experiment, *wildtype* mice (not expressing hCD155 and, thus, not permissive for poliovirus infection) were immunized with mOVA2-infected FLT3L-BMDCs (generated from hCD155-tg mice).

Though NK cells are capable of clearing virally infected DCs, it is not a certainty that all virus-infected dendritic cells are recognized and destroyed by NK cells. Indeed, our data provide compelling evidence that dendritic cells infected with PVSRIPO-based vectors are not eliminated by the host and are capable of priming antigen-specific T cells *in vivo*. For example, our data in new **Supplementary Figure 15** show that mOVA2-infected dendritic cells produce SIINFEKL-specific CD8 T cell responses *in vivo*. Lastly, the primate study in Shen *et al.* showed intact, (presumably) live poliovirus antigen-positive CD11c+ gastrointestinal myeloid cells *eight and ten days post oral virus challenge* with wild type poliovirus. This suggests that, even in natural infections of primates with wild type polioviruses, infected myeloid cells are not targeted for destruction by host immune cells;

thus, virus targeting of DCs does not preclude DC-mediated T cell priming *in vivo*.

“3) The authors do provide evidence that inducing a T cell response against OVA can repress a melanoma cell line expressing the OVA-antigen. Albeit the question remain whether this approach can increase rejection of a less immunogenic cancer, such as GBM or DIPG. Further, GBM patients are known to have a more exhausted T cell response and thus might be much less responsive towards the treatment. Whether a single-antigen expressed by the poliovirus in a GBM or DIPG setting, as currently suggested, is able to be of therapeutic value is questionable.”

We fully agree with the reviewer that targeting single tumor antigens, i.e. with cancer vaccines, is not ideal for combatting aggressive, invasive neoplasias such as malignant gliomas. Conceptually, we prefer strategies that can deliver intense, broad immune engagement of cancers and are agnostic to the notorious heterogeneity of advanced neoplasia. We successfully use intracerebral inoculation of PVSRIPO in recurrent glioblastoma with these objectives.¹¹

However, an approach with intratumoral inoculation of PVSRIPO may not be feasible in tumors growing in the brainstem, due to the local peritumoral inflammatory effects that occur in the brain (please see Desjardins *et al.*¹¹ for a detailed discussion of this phenomenon in malignant glioma patients in the clinic). This type of inflammation, which is desirable from an immunotherapy standpoint due to its broad immune-engaging effects,³ can be controlled in supratentorial tumors. However, DIPGs are located in the brainstem, a vital structure that may not tolerate peritumoral inflammatory responses to PVSRIPO infection without deleterious neuroinflammatory sequelae.

Therefore, immunotherapy options with tumors growing in the brainstem are inherently limited. In that context, the H3.3^{K27M} mutation provides a rare and unique opportunity: (i) it is a driver oncogenic signature homogeneously expressed in all tumor cells; (ii) it is categorically tumor-specific, and it is a validated HLA-A2 epitope, shown to produce CD8 T cell responses in patients;¹ (iii) there is a great need for better therapy options for DIPG, a uniformly lethal diagnosis with dismal prognosis that is associated with grievous suffering for patients and their families.

We believe that the fact that the H3.3^{K27M} signature is a driver oncogenic signature of DIPG homogeneously expressed in all tumor cells provides a rational foundation for our approach. As the Reviewer notes, the notorious heterogeneity of most advanced therapy-resistant cancers would likely limit the efficacy of vaccination against single tumor antigens.

That said, our approach can be used to target several different tumor epitopes simultaneously; this can be achieved by constructing diverse vectors expressing different epitopes and immunizing with a ‘vector cocktails’; such an approach was taken by Andino *et al.* for broadly targeting HIV epitopes.¹⁴ We now show additional evidence in our revised submission that our vector platform can be flexibly adapted to accommodate any desirable

foreign sequence for delivering a wide range of epitopes (see responses to **Reviewer 2**, pg. 17-18; to **comment #2** of **Reviewer 4**, pg. 29-30; and **new Supplementary Figure 3**).

“4) Vaccination strategies often include prime/boost administration. Adaptive immunity against the poliovirus in the prime phase might decrease its efficacy as an antigen-deliver vector.”

The reviewer is right. Many studies have outlined the effect of adaptive anti-viral (vector) immunity as a key factor in limiting vector immunization efficacy. A role for pre-existing immunity in shaping the host immune response is likely in our case, since prospective patients would be seropositive for anti-poliovirus type 1 neutralizing antibodies (expected to be an enrollment criterion, similar to PVSRIPO).¹¹

Poliovirus (or derivative vector) infections are naturally acute and short-lived, because these +strand RNA viruses cannot persist, induce chronicity or latency. Enteroviruses (polio is their flagship member) overcome this obstacle with speed. Picornavirus/enterovirus infections are extremely rapid and efficient and geared towards achieving peak viral propagation prior to the host immune system clearing the infection (*in vitro*, mature viral progeny is released from infected cells as early as four hours post receptor binding). Thus, delivering a high-dose vector inoculum directly to the site of (limited) propagation, e.g. by intramuscular inoculation, will elicit local vector propagation in the presence of preexisting antiviral immunity.

This has been documented empirically before by Mandl *et al.*¹⁵ with a (genetically instable) poliovirus vector design. In their study, Mandl *et al.* showed that immunization of hCD155-tg mice with a wild type poliovirus-based vector delivering SIINFEKL achieved similar efficacious humoral and CTL responses in pre-immunized and naïve animals. A modest (~20%) decrease in the CTL response of pre-immunized mice was abrogated by increasing the vector dose from 10^6 pfu to 2×10^7 pfu (by intraperitoneal immunization), which is a reasonable dose for preclinical tests of a poliovirus construct *in vivo*.¹⁵

We do not expect PVSRIPO-based vectors to be capable of setting up extended periods of propagation in target tissues (e.g. skeletal muscle), nor do we believe that such a scenario would be desirable. As is implicit in Mandl *et al.*,¹⁵ acute local vector propagation at the skeletal muscle immunization site –giving rise to local dendritic cell infection— similar to the natural scenario in the primate gastrointestinal tract,⁷ occurs in the presence of preexisting anti-poliovirus immunity.

We do not advocate this with our approach, but the issue with pre-existing immunity has been addressed with a heterologous prime-boost strategy; e.g. prime with vector and boost with peptide plus adjuvant. Such heterologous prime-boost regimens were used with vaccinia virus-based vectors, whose immunization efficacy was shown to be highly sensitive to pre-existing immunity (see discussion in Mandl *et al.*).¹⁵ Thus, vaccinia vector-mediated immunization benefits from heterologous prime-boost in HIV vaccine regimens.¹⁶

“5) Most of the validation is carried out the OVA antigen, a very strong foreign

antigen. It remains debatable whether responses against weaker tumor antigens, such as DIPGs, will be as effective.”

The reviewer is right – OVA is notorious for its high potency as an antigen. Most tumor-associated/specific antigens are inherently weak epitopes.

The main reason for inclusion of the SIINFEKL antigen in our studies is that it has model character, is exceedingly well studied, has many dedicated research resources (for assessing immune responses), and permits comparison of our vector approach with competing strategies, e.g. SIINFEKL peptide antigen + poly(I:C) adjuvant. Since the concept of PVSRIPO as an immunization vector is new, it is highly advisable to carry out proof-of-principle studies in a standardized model, such as SIINFEKL/OVA (see also our response to **comment #4** of **Reviewer 3**, pg. 24-26).

A second reason for the inclusion of SIINFEKL is that it binds strongly to the C57Bl6 MHC Class I (H2Kb). Many human epitopes of therapeutic interest, including H3.3^{K27M}, have poor intrinsic binding affinity for mouse MHC and, thus, do not lend themselves for studies in rodent tumor models. Carrying out investigations with SIINFEKL immunization in the B16F10.9-OVA model permits proof-of-principle tests of PVSRIPO-based vectors without this MHC mismatch issues.

We have made a substantial effort to develop a new mouse model for assessing immunization against the H3.3^{K27M} epitope in an orthotopic malignant glioma model, by generating murine glioma cells and new mouse strains transgenic for an established, engineered version of HLA-A2, capable of binding the H3.3^{K27M} epitope. Please refer to our detailed responses to the **Editor** on this topic (items 1.-5.; pg. 1-3).

Lastly, the fact that most tumor antigens are inherently weak, is precisely the reason why we are pursuing PVSRIPO as a vector platform. Unlike all other RNA viruses, whose 5'ppp-viral RNAs are recognized by RIG-I, PVSRIPO naturally engages the MDA5 pattern recognition receptor and its extended host RNA surveillance network.¹⁷ Picornaviruses carry a 5' genome-linked protein (Vpg), which precludes detection by RIG-I.

Groundbreaking investigations of MDA5's central role in detection of- and coordinating the innate response to endogenous dsRNAs, has implicated MDA5 in breaches of self-tolerance/autoimmunity.¹⁸ For these reasons, the specific inflammatory footprint of PVSRIPO infection in dendritic cells that we documented in this work, coordinated by the MDA5 nexus, may be particularly suitable for immunization against inherently weak human tumor antigens. In this context, it is of utmost importance that both, expression of the tumor antigen and the MDA5-coordinated innate response occur in the *same, infected antigen presenting cell*. See our general response and statement of premise above (pg. 5, 6).

“6) Figure 5: ELISPOT and IFN-gamma staining lack a positive control, providing additional information that the potential to secrete IFN-gamma is maintained and percentage of IFN-gamma producing cells is similar over all samples.”

The reviewer is right. The assay reported in our original manuscript was performed with a positive control (as is standard practice for ELISPOTs in our lab), splenocytes treated with concanavalin A (activator of T cells).

We now show the data from the positive control in our assay in **new Supplementary Figure 13**.

“7) Figure 6: An Untreated control is lacking from the figure. This tumor rejection model is modelling melanoma and does not provide information concerning a more immunogenic cold tumor, such as GBM or DIPG, nor does it provide information whether this immune response will be limited by the blood-brain barrier.”

Please see also our comments to **Reviewer #3 comment# 4** (pg. 24-26).

Since PVSRIPO-based vectors induce locoregional inflammation that could influence tumor progression (see **Figure 4a-c**), the appropriate control for mOVA2/mRIPO(H3.3) immunization is vaccination with empty vector (mRIPO δ 6) (this was reported in **Figure 4**). mRIPO δ 6 will produce the same locoregional inflammatory response as mOVA2/mRIPO(H3.3) immunization without expression of the immunogenic epitope.

However, we did perform an mOVA2 immunization experiment in the B16 murine model that included a mock-treated control group. This experiment was not reported in the original manuscript. We have now added this data in **new Supplementary Figure 17**.

While it is true that melanoma frequently exhibits signs of intrinsic immune-engagement (immunologically ‘hot’ tumor) *in patients*, this is distinctively not the case for the B16 murine melanoma model. There is much empirical evidence for this, including a thorough, systematic study of commonly used mouse tumor models, where B16 was identified as one of the ‘coldest’, least immune-engaged tumors when compared to other commonly used murine tumor models.¹⁹

In contrast, available syngeneic murine gliomas, e.g. the commonly used SMA560 (VMDk), GL261 or CT2A (both C57Bl6), are characterized by relatively high intrinsic immune engagement, as is evident by substantial amounts of tumor-infiltrating T cells in orthotopic tumor implants. This scenario is decidedly not representative of the clinical situation, where malignant gliomas are notoriously devoid of tumor-infiltrating lymphocytes.

Thus, the spectrum of ‘cold vs. hot’ tumors described for patient cancers, e.g. malignant gliomas vs. melanomas, is not reflected in immunocompetent mouse tumor models used to represent them.

We have developed a new mouse model for assessing immunization against the H3.3^{K27M} epitope in an *orthotopic malignant glioma model*. Please refer to our detailed responses to the **Editor** on this topic (items 1.-5.; pg. 1-3).

There is reappraisal of the old dictum of the immune-privilege of the CNS, based on much recent evidence that immune cells (including CD8 T cells) can pass through the blood brain barrier, particularly during inflammation.^{20, 21}

“8) The evidence connecting T cell priming and polio is not that strong. Polio does not stimulate (or mature) antigen presenting DC as IL12 is negative (in Fig S3 and Fig 3) while the IFN pathway appears to be stimulated. This implies that pDC, cDC or macrophages are most likely the target of Polio. I also question the FACS analysis, where the gating method seems inconsistent through experiments with even some errors (7AAC positive = dead, but same figure S4 says live cells).”

We disagree with the Reviewer on this point:

First, while IL12 release was not evident in (mouse) GMCSF- and FLT3L-induced BMDCs infected with mOVA2 *in vitro* (**Figure 3**), it did occur in human GMCSF-differentiated dendritic cells infected with RIPO(H3.3) (**Figure 7**).

Second, it is important to note that IL12 production in dendritic cells is potently stimulated by CD40:CD40L interaction [Cella, M. *et al.* Ligation of CD40 on dendritic cells triggers production of high levels of interleukin-12 and enhances T cell stimulatory capacity: T-T help via APC activation. *J Exp Med* **184**, 747-752 (1996); Quezada, S.A. *et al.* Mechanisms of donor-specific transfusion tolerance: preemptive induction of clonal T-cell exhaustion via indirect presentation. *Blood* **102**, 1920-1926 (2003)].

Also, seminal work established that for maximal IL12 production induced by pathogen-associated molecular patterns (as in our case), dendritic cells need the CD40L (CD154) signal [Schulz, O. *et al.* CD40 triggering of heterodimeric IL-12 p70 production by dendritic cells *in vivo* requires a microbial priming signal. *Immunity* **13**, 453-462 (2000)].

Our assays with mOVA2/RIPO(H3.3) infection of dendritic cells were performed with isolated dendritic cells *in vitro*; for maximal induction of IL-12, the CD154/CD40L signal from T cells is required. Thus, higher levels of IL12 production upon PVSRIPO challenge should be observed *in vivo*, in the presence of T cells.

This is exactly what our data show. Cytokine analyses of skeletal muscle from mOVA2-challenged mice revealed local IL12 release (**Figure 4a**). This occurred in the context of potent CD40 induction by PVSRIPO in locoregional dendritic cells in skeletal muscle (**Figure 4c**). Thus, our findings corroborate the findings by Schulz et al. (2000) (and other literature), implicating innate antiviral responses (in dendritic cells), upregulation of the CD40 maturation marker in dendritic cells, engagement of CD40:CD40L interactions *in vivo* (in the presence of T cells), in a complex physiological system involving multiple compartments that control IL12 release.

Third, we carefully designed our gating strategies with positive and negative controls and performed isotype staining whenever necessary. Although unclear, we assume that the reviewer's comment refers to the Original Supplementary Figure 4b. We now provide additional controls for the staining: isotype control staining and BMDCs treated with LPS

(positive control) in **revised Supplementary Figure 8**. We are not aware of any inconsistencies in our gating strategies and kindly ask the reviewer to identify these instances so we can correct them.

Fourth, the Reviewer is right in pointing out ambiguous labeling in the Original Supplementary Figure 4. Our gating strategy described in the Original Supplementary Figure 4 showed a panel, which might have suggested that we had gated on dead cells (7AAD+ cells) (**Rebuttal Figure 1**). Obviously, this was not the case: the cells 7AAD+ cells were gated *out*; we agree, however, that the '7AAD' label was misleading.

Rebuttal Figure 1: original version of '7AAD' panel in original Supplementary Figure 4.

We have corrected the label for this panel, which now reads 'gate-out dead cells' (**Rebuttal Figure 2**).

Rebuttal Figure 2: amended version of panel in new **Supplementary Figure 7**.

"9) I question the vaccine strategy of Fig5. IM immunization requires draining Lymph Nodes for proper T cell population stimulation. OVA-T cells in splenocyte and blood suggests mOVA2 viruses are leaked systemically as T cell activation in the control polyIC:SIINFEKL was negligible, but should be more positively activated."

We respectfully disagree with the comments about systemic spread of mOVA2 and its potential implications for the ensuing host immune response.

First, we now show evidence for cells expressing H3.3^{K27M} in (popliteal) lymph nodes draining the (gastrocnemius muscle) immunization site (see our response to **comments #1** and **#2**, pg. 7-8).

Second, we have provided compelling evidence in mOVA2-immunized *hCD155*-tg mice for (modest) locoregional virus replication in skeletal muscle (in line what is known for poliovirus replication in skeletal muscle of *hCD155*-tg mice; **new Supplementary Figure 9**), a potent host locoregional inflammatory response characterized by profuse proinflammatory cytokine release (**Figure 4a**), local recruitment of leukocytes, incl. CD4/CD8 T cells, as well as CD11c+ and CD11b+ myeloid cells (**Figure 4b**), and induction of maturation markers in antigen presenting cells in skeletal muscle tissue (**Figure 4c**). Therefore, there is strong empirical evidence for a locoregional response at the immunization site in skeletal muscle, setting the stage for dendritic cell homing to the draining lymph nodes, presentation of the H3.3^{K27M} epitope to T cells, and a systemic CD8 T cell response.

It is likely that the mOVA2 inoculum does not remain 100% contained at the immunization site. Systemic dissemination, invariably associated with dilution of the agent, is unlikely to contribute to SIINFEKL-specific CD8 T cell responses, especially given the extremely restrictive poliovirus tropism (see our response to comments #1 and #2, pg. 7-8).

We ascribe the superior T cell response to mOVA2 –compared to SIINFEKL + poly(I:C)—to the fact that PVSRIPO-based vectors deliver the antigen *and* powerful, stimulation (co-stimulatory molecule upregulation, pro-inflammatory cytokine, e.g. type I/III IFN, production) *to the same cell*. This is supported by our companion control showing that the combination of (empty) mRIPO δ 6 vector plus SIINFEKL peptide also delivers an inferior response. Likewise, co-delivering poly(I:C) with SIINFEKL peptide likely will not target the same cell(s) and will result in out-of-sync antigen uptake and proinflammatory stimulation (**Figure 5d**).

The benefits of such co-targeting (epitope presentation and activation of the same antigen presenting cells) have been documented previously with conjugating HIV-1 Gag with TLR7/8 agonist, which was shown to induce much stronger responses than immunization with Gag polypeptide mixed with TLR7/8 agonist.²²

“10) The response of Figure 6B is not durable and not that impressive, again questioning vaccine efficacy.”

We disagree with this assessment.

The record for survival data obtained with preclinical tests of experimental anti-cancer agents in immunocompetent mouse tumor models for predicting therapy success in patients is dismal.

Just one example of many:

A ‘murinized’ form of Bevacizumab (anti-VEGF antibody) achieved modest extension of survival in the immunocompetent GL261 malignant model (median survival was extended from 20.0 to 27.5 days).²³ Durable responses/prolonged survival were not observed. In a different study in the same model, immune checkpoint blockade with murinized anti-PD1 achieved impressive results with disease eradication (‘cures’) in 56% of treated mice.²⁴

Yet, clinical studies of nivolumab (anti-PD1) in recurrent GBM failed due to futility (<https://news.bms.com/press-release/corporatefinancial-news/bristol-myers-squibb-announces-phase-3-checkmate-498-study-did>). In contrast, Bevacizumab has obtained full approval for the same indication.

In our view, mouse models have an important role to play in testing key hypotheses, which are based on mechanistic experimentation in tissue culture, in an *in vivo* setting. This is what our empirical animal work has accomplished: we have obtained compelling data that (i) PVSRIPO vector immunization elicits locoregional inflammatory stimulation, immune cell (e.g. dendritic cell) infiltration and local induction of dendritic cell maturation markers

in vivo; (ii) PVSRIPO-based vectors target host dendritic cells for infection and foreign epitope expression *in vivo* (**new Figure 4d**); (iii) PVSRIPO vector-infected dendritic cells home to lymph nodes draining the immunization site *in vivo* (**new Figure 4d**); (iv) they elicit SIINFEKL- or H3.3^{K27M}-specific CD8 T cell responses *in vivo* (**Figure 5c, d**; **new Figure 5e, f**); (v) they induce immune cell infiltration and CD8 T cell recruitment into distant tumors *in vivo* (**Figure 6**); and (vi) they oppose tumor progression in immunocompetent subcutaneous and intracerebral rodent tumor models *in vivo* (**Figure 6**; **new Supplementary Figure 17**; **new Figure 7**; **new Supplementary Figure 18**).

There are a myriad of reasons why therapy success in immunocompetent rodent models is not predictive of the clinical response in patients. Most commonly used immunocompetent rodent models are highly aggressive, clonal cells lines with a history of high-passage *in vitro/in vivo* selection. Both models in our study kill animals (CT2A \geq 15% weight loss, B16F10.9-OVA tumor \geq 1000mm³) rapidly after tumor implantation. They do not properly recapitulate key aspects of the tumor microenvironment in human cancers, such as relations with tumor-associated macrophages, the extent and composition of intrinsic T cell infiltration, etc.²⁵

Moreover, PVSRIPO-based vectors are exquisitely human-specific, because both parent viruses its genome is composed of, poliovirus type 1 (Sabin) and human rhinovirus type 2, are exclusively human pathogens. We use mouse-adapted vector variants [mOVA2, mRIPO(H3.3)] in our investigations in rodent tumor models *in vivo*. Yet, *the* decisive mechanistic element of PVSRIPO-based vectors, the virus:host(innate) immune interface, based on 1,000s of years of enterovirus co-speciation and co-evolution with *homo sapiens*, cannot be appropriately modeled in non-human hosts. This is evident in the data reported in our manuscript; for example, intrinsic permissiveness for PVSRIPO vector translation and propagation is higher in human GMCSF-derived PBMCs (**Figure 7a**) than in their murine counterparts (**Figure 3a**).

Reviewer #2, expert in oncolytic virus design (Remarks to the Author):

These authors have published extensively on the oncolytic virus poliovirus:rhinovirus chimera PVSRIPO. This virus has demonstrated promise in clinical and preclinical studies. In this manuscript, the authors report on modification the poliovirus:rhinovirus chimera PVSRIPO in the stem loop region to promote the stable expression of exogenous antigens, while preserving the pre-existing attributes for this vector including the following:

- 1. tropism for antigen presenting cells*
- 2. no interference with immune activation (unlike many viruses with immune avoidance genes)*
- 3. antigen presentation*
- 4. robust engagement of inflammatory responses*

The authors convincingly demonstrate that these stem loop modifications did not adversely affect the intended translation initiation, the foreign transgene insert was expressed and the product was processed properly.

In addition, the authors designed and implemented various in vitro and in vivo experiments to demonstrate that type I IFN was induced by the engineered virus, antigen presentation occurred, and that tumor antigen-specific CD8 T cells were induced and activated by treatment with the antigen-expressing virus. Finally, the authors demonstrate that CD8 T cells migrated into the tumor site, and that this infiltration was associated with a reduction in tumor burden and an increase in animal survival in one mouse model.

“Generalizability: Given the limited scope of tumor models explored, and limited foreign antigen transgene constructs evaluated, the likely generalizability of this approach to other systems and antigens cannot be assessed.”

The reviewer is right, and our claims of generalizability were not properly buttressed by empirical evidence.

Our approach is a platform technology that can be adapted to accommodate any foreign sequence, provided that the strict limitations on insert size and structure mandated by the genetic stability dictum are adhered to. To support this argument, we now included the structure of five additional designs [one targeting the IDH2(R172G) signature of malignant gliomas, the other delivering various signatures derived of the SIV p55 Gag protein] (**new Supplementary Figure 3**). These additional vector prototypes had been constructed and tested prior to submission of the original manuscript.

The five vector prototypes were constructed according to the same design principles applied to mOVA2 and RIPO(H3.3) (see structures in **new Supplementary Figure 3**), and were characterized with regard to genetic stability in the same manner as the mOVA2 and RIPO(3.3) vectors. All constructs shown in **new Supplementary Figure 3** retained the foreign insert without acquiring adaptation mutations according to the criteria laid out in our manuscript.

The structural/functional significance of the 100% conserved cryptic AUG at the base of stem loop domain VI, stem loop domain VI itself, and the ‘spacer’ (the region spanning from the cryptic AUG to the actual initiation codon) in enteroviral 5' untranslated regions is not understood. Therefore, the defining features of a genetically stable foreign insert cannot be rationally predicted with our approach. We thus rely on functional assessments, such as serial passage (as performed in assays described in **Figure 1**), and expression analyses by immunoblot (see **Figure 2**).

We cannot exclude the possible existence of specific sequences (encoding for desirable target epitopes) that may not be suitable for our strategy. Every vector design attempted in our laboratory led to successful derivation of a stable prototype. Thus, the added evidence provided in our revised manuscript supports our claims of generalizability.

Technical questions & comments:

“While the authors have shown that there was genetic stability preserved in the foreign insert after 20 passages of virus in cells in culture, they did not comment on the stability in the rest of the viral genome. Did they sequence the whole viral genome to rule out any potential change in other regions of the genome that were not around the locus of the foreign insert? This could have significant safety implications.”

We have >25 years of experience with the problem of genetic stability with poliovirus recombinants intended for clinical use, stemming from our work with the vector parent, PVSRIPO.¹¹

The backbone for RIPO(H3.3) is the live attenuated, type 1 poliovirus (Sabin) vaccine (PV1S). PV1S was/is safely given to many billions of healthy infants as polio prophylaxis in the past 60 years. There are rare instances of reversion to neurovirulence with the live attenuated poliovirus vaccines (VAPP; vaccine-associated paralytic poliomyelitis). Of the six (principal attenuating) point mutations in the PV1S genome with proposed roles in the neuro-attenuation phenotype of PV1S (based on laboratory tests),²⁶ mainly the critical A480G attenuating substitution (in the internal ribosomal entry site, IRES) is linked to VAPP in human vaccine recipients.

PVSRIPO/or PVSRIPO-derived vectors do not contain the PV1S IRES or its A480G substitution, because the entire IRES is derived of human rhinovirus type 2. Since there are no obvious targets for monitoring the genetic stability in the PV1S backbone of PVSRIPO-based vectors, there is no compelling reason to perform full-genome analyses of genetic stability.

The genetic region mediating neuro-attenuation in PVSRIPO –and in PVSRIPO-based vectors— is the heterologous human rhinovirus type 2 (HRV2) IRES, which imparts a far more pronounced and genetically stable neuro-attenuation phenotype than its PV1S counterpart (the PV1S IRES with its A480G substitution). This is exceedingly well

documented in the literature, and we refer the Reviewer to the published empirical record on the topic.^{8,9,27-31}

For these reasons, our efforts to define genetic stability with PVSRIPO-based vectors are focused on the viral 5' UTR, encompassing the HRV2 IRES and the foreign insert.

We have extensively documented genetic stability, and the mechanisms of neuro-attenuation of the foreign HRV2 IRES in PVSRIPO, in prior publications reporting on non-human primate IND-directed toxicology for PVSRIPO⁸ and in IND-driven studies of genetic stability in the intended target *in vivo*.²⁷

“The authors should also consider evaluating this system in additional tumor, and with additional antigen transgene constructs to demonstrate the likely generalizability of this approach to other systems and antigens.”

The reviewer is right and we have amended our manuscript following his/her suggestions. Our strategy is a platform technology, which can be flexibly adapted for immunization of any target antigenic signature, provided the empirically established limitations on insert size and structure are respected.

We added a range of other PVSRIPO-based vector designs that we have generated, and that were successfully tested for genetic stability (see **new Supplementary Figure 3**). This clarifies our assertion that our vector strategy is applicable to functional integration of virtually any foreign insert into the IRES.

We generated a new mouse model for modeling H3.3^{K27M}+ malignant glioma with orthotopic, intracerebral implantation in AAD_hCD155-tg mice. Please see a detailed description of this approach in our response to the **Editor**; items 1.-5., pg. 1-3.

“In addition, it is likely that the oncolytic effect of this virus will be reduced by increasing its immunogenicity. This question should be addressed by evaluating the relative efficacy of a tumor antigen expressing PVSRIPO in a tumor model demonstrating potent oncolytic virus replication and efficacy.”

We have not tested PVSRIPO vectors for usage as oncolytic agents here, but rather as vaccine vectors, since the vaccination site is distant from the target (tumor).

We assume that this comment refers to the potential of host effector anti-H3.3^{K27M} responses for eliminating RIPO(H3.3)-infected host cells (e.g. dendritic cells), thereby limiting the vaccine vector's efficacy. While this scenario is possible, in principle, we believe that our data exclude such an event. Please also see our detailed responses to **comments #1** and **#2 (Reviewer 1, pg. 7-8)** about the possibility of host immune cell-mediated elimination of RIPO(H3.3)-infected dendritic cells.

Poliovirus itself is a potent immunogen; RIPO(H3.3)-infected host cells will also express poliovirus antigens and, thus, be subject to the possibility of immune recognition and

elimination. We do not believe that expression of the H3.3^{K27M} epitope in vector-infected cells will enhance the likelihood of this to occur.

For a detailed discussion of the short-lived, acute nature of host:poliovirus interactions, the absence of viral persistence, chronicity or latency, and the role of pre-existing anti-poliovirus immunity in shaping these interactions, we kindly refer the Reviewer to our detailed response to comment #4 of Reviewer 1, pg. 10.

Reviewer #3, expert in clinical glioma immunotherapy (Remarks to the Author):

In this manuscript, Mubeen et al. designed a poliovirus/rhinovirus chimera-based vector, named mOVA2, which infects dendritic cells(DCs) such that antigen peptide (epitope) carried by the virus can be expressed by DCs, followed by antigen presentation to cytotoxic CD8+ T cells. mOVA2 overcomes the neuropathogenicity and genetic instability originated from the poliovirus, while maintaining its tropism to DCs.

The authors demonstrated that mOVA2 infects, activates and induces epitope presentation in mouse bone marrow derived dendritic cells (BMDCs) and the following activation of CD8+ T cells in vitro. In addition, mOVA2 recruits and activates DCs, and triggers antigen-specific T cell activation in vivo. Using a murine model of melanoma, they showed that mOVA2 recruits CD8+ T cells to the tumor, reduces tumor size and increases survival.

As a proof of concept experiment, Mubeen et al. engineered the virus [named RIPO(H3.3)] to express epitope containing H3.3-H27M, a signature mutation found in patients with diffuse intrinsic pontine glioma (DIPG). The authors showed that RIPO(H3.3) activates human BMDCs and triggers responses from H3.3-K27M-TCR+, CD8+ Jurkat cells. RIPO(H3.3) has a good translational value given that intratumoral infusion of PVSRIPO virus in patients with recurrent grade IV malignant glioma confirmed absence of neurovirulence and increased survival in past studies.

The study is of great potential clinical and scientific significance. However, the writing is so filled with jargon and lacking in crucial details that it is very difficult to follow the manuscript. The main concern is that the authors need to experimentally validate that this viro-immunotherapy shows preclinical efficacy in a valid experimental model of DIPG in vivo. I am unclear what the B16F10.9-OVA tumor model really is, but I do not believe it is a DIPG model.

And the manuscript needs to be edited substantially; I refer the authors to the following guide on scientific writing: http://www.georgegopen.com/uploads/1/0/9/0/109073507/gopen_swam_sci_of_sci_writing_am_sci_1990.pdf.

Please see the following specific suggestions:

“1. Fig. 2C: a lysate immunoblot showing the total Flag-tagged proteins should be provided for the IP experiment. In addition, what are the reasons for disappearance of the three flag-tagged bands at 6 and 8 hpi?”

The reviewer brings up important issues, as our vector design involves engineered processing of viral precursor proteins. This design requires rigorous, in-depth accounting for the expression of viral polypeptides induced by vector infection of host cells.

Our laboratory has much experience with expression of Flag-tagged proteins followed by Flag-immunoprecipitation for empirical studies.³²⁻³⁶ Available anti-Flag antibodies have very high background activity in immunoblot of cell lysates, essentially precluding rational analyses of Flag-tagged proteins using this method.

However, Flag-tagged proteins can be very efficiently and specifically isolated by elution with Flag-conjugated beads. This is the method employed in our study in **Figure 2c**. The raw-data immunoblot with Flag-antibodies and (viral) 2C antibodies of the samples analyzed in **Figure 2c** is shown below (**Rebuttal Figure 3**). In the left-hand panel, a multitude of bands represent non-specific signal picked up with the anti-Flag antibody (we have observed this type of non-specific staining in similar assays of cell lysates countless times before). The fact that the banding pattern is similar in the 0 and 4 hours post infection lanes, indicate that Flag immunoblot cannot specifically detect any Flag-tagged viral products due to very high intrinsic non-specific background staining.

The right-hand panel shows the corresponding anti-2C immunoblot (compare to the same assay in a different run in **Figure 2b**). Both filters were deliberately overexposed to pick up all detectable signal:

Rebuttal Figure 3. Immunoblot analyses of the immunoprecipitation samples shown in **Figure 2b** in the main Manuscript.

We do not think that showing the image of the Flag immunoblot is informative, but will add it to our Manuscript if the Reviewer/Editor deems this information essential.

The disappearance of the viral Flag-polypeptides in the assay shown in **Figure 2b** is due to (intended) proteolytic cleavage of viral precursor polyproteins. The viral Flag-fusion polypeptides evident in our assay are subject to cleavage by the viral 2A protease, because the foreign sequence insert (including the Flag-tag) is separated from the coding region for the viral polyprotein by an engineered 2A^{pro} cleavage site (see text for detailed description of our vector design; details in **Figure 1**). Therefore, only viral polypeptides that retain the N-terminal Flag prior to 2A^{pro}-directed cleavage (i.e. at 4 hours post infection), can be detected. As these viral polypeptides are proteolytically processed by 2A^{pro}, which occurs

rapidly after infection, they no longer can be isolated and detected by Flag-IP at infection intervals after 4 hours post infection.

The same phenomenon is evident with RIPO(H3.3), where H3.3^{K27M}-specific signal peaks at 3h post infection and decreases thereafter (**Figure 2e**). However, detection of the H3.3^{K27M}-specific signal is much more efficient and occurs for extended periods despite 2A^{pro} processing, because a commercially available H3.3^{K27M}-specific antibody (Cell Signaling; D3B5T) has relatively low background activity in immunoblot and can be used to detect the antigen directly (see product information on the Cell Signaling website), without the need for immunoprecipitation (see **Figure 2e**). Thus, immunoblot detection of H3.3^{K27M} is possible for a more prolonged interval than with the Flag-elution method employed for demonstrating Flag-SIINFEKL expression. This is also explained in our Manuscript; see **Results**, pg. 7-8.

“2. Fig. 3E: what are the results when the same experiments are done in FLT3L BMDCs?”

This is a very good suggestion, as FLT3L-differentiated BMDCs (a model for representing Batf3+ dendritic cells) have been proposed to assume indispensable functions in classical- and cross-presentation of antigens and, thus, may play an important role with our approach. Also, PVSRIPO exhibited an intriguingly distinct phenotype in FLT3L-differentiated BMDCs, characterized by delayed innate antiviral defenses, enhanced permissiveness for viral translation, and superior type I/III IFN-dominant proinflammatory activation (**Figure 3**).

We believe that our analyses of OT-I co-culture with GMCSF-differentiated BMDCs yielded compelling evidence for mOVA2-infection of dendritic cells to induce SIINFEKL presentation to T cells (**Figure 3e**). Based on thorough, comprehensive evaluation of the FLT3L-differentiated BMDC phenotype after mOVA2 infection (**Figure 3e**), it is reasonable to assume that efficient SIINFEKL antigen presentation would occur with these cells as well. Accordingly, when FLT3L-BMDCs were infected with mOVA2 as a ‘dendritic cell vaccine’ approach (**new Supplementary Figure 15**; see our response to **comments #1** and **#2**; **Reviewer 1**, pg. 8) we observed the generation of SIINFEKL-specific CD8 T cells. This indicates that mOVA2-infected BMDCs also present SIINFEKL.

Therefore, since our assay was designed primarily to establish the principle of antigen presentation after PVSRIPO vector infection of dendritic cells, we do not feel that it is necessary to conduct the same assay with FLT3L-differentiated BMDCs.

“3. In addition to eIF4G1 cleavage, are proliferation, cell death, and senescence affected in the infected BMDCs?”

The reviewer brings up an excellent point. The central premise of our approach is that PVSRIPO-based vectors will elicit sub-lethal, protracted viral translation/replication in dendritic cells, without interfering with the antigen-presenting and T cell costimulatory

functions of such cells. Obviously, any detrimental effects of PVSRIPO vector infection on dendritic cell well-being would counter this desired outcome.

We have accumulated overwhelming evidence supporting our premise, by conducting thorough dendritic cell infection studies in a range of dendritic cell cultures derived of mice and humans: murine (GMCSF- and FLT3L-differentiated BMDCs) (**Figure 3a, b**) and human (GMCSF-differentiated) (**Figure 7a**) dendritic cells. Infection of these cells invariably leads to subdued and protracted viral translation (**Figure 3a, b, Figure 7a**), limited viral replication (**Figure 7b**), and a robust innate antiviral host response that is likely responsible for containing vector translation and replication (**Figure 3a, b, Figure 7a**).

We have shown that PVSRIPO vector-infected dendritic cells: (i) exhibit induction of multiple key maturation markers; (ii) mount a powerful, intrinsic type I/III IFN-dominant response in infected dendritic cells; (iii) profusely release a range of proinflammatory cytokines; (iv) present vector-delivered antigens to T cells; (v) provide the correct costimulatory context for activating CD8 T cell responses *in vitro* and *in vivo*; (vi) migrate to lymph nodes draining the immunization site *in vivo* (**new Figure 4d**).

This solid evidence for broadly *enhanced* function of dendritic cells after PVSRIPO vector infection *in vitro* and *in vivo* all but exclude the possibility of cell death or senescence induced by the presence of actively translating and propagating vector inside dendritic cells. GMCSF/FLT3L-differentiated dendritic cells are terminally differentiated and do not proliferate.

Also, we now include new evidence with an mOVA2-infected dendritic cell vaccine (suggested by **Reviewer 1**; see **comments #1** and **#2**, pg. 8) in *hCD155*-tg mice (**new Supplementary Figure 15**). This assay demonstrates that adoptive transfer of (*ex vivo*) mOVA2-infected, GMCSF-differentiated BMDCs produces SIINFEKL-specific CD8 T cell responses. This result further emphasizes that PVSRIPO vector-based infection of dendritic cells does not interfere with their viability or immune function.

A key distinction of PVSRIPO, compared to its type 1 (Sabin) vaccine precursor (or to wild type, type 1 polioviruses), is its sub-lethal, non-cytotoxic phenotype in human myeloid cells. This is due to the heterologous rhinovirus IRES element in PVSRIPO. We have documented this phenomenon for PVSRIPO and addressed dendritic cell viability, its mechanisms, and its functional implications in a prior publication.³

“4. There is no reference or introduction for the B16F10.9-OVA melanoma tumor model. How is this relevant to H3.3K27M+ glioma?”

The Reviewer is right. We added an introduction of the B16F10.9-OVA model to the text, to better inform the reader of the purpose of our empirical strategy. See **revised Results**, pg. 14. Please also see our response to a similar comment from **Reviewer 1** (**comment #5**, pg. 10-11).

The concept of immunization with PVSRIPO-based vectors is entirely new and, thus, categorically requires rigorous proof-of-concept validation in standardized models where appropriate research tools are available and where comparisons to competing approaches (e.g. H3.3^{K27M} peptide vaccines + adjuvant) are possible.

There currently is no accepted standard rodent model for assessing immunotherapy in H3.3^{K27M}-positive glioma/DIPG. Thus, evidence obtained in any new mouse model (such as the one we derived for our investigations, see our responses to the Editor; items 1. - 5.; pg. 1-3) must be buttressed by parallel investigation in a standardized system that permits the use of established research tools to evaluate immune responses against standard antigens (see below).

The H3.3^{K27M} target epitope strongly binds to human MHC I (HLA-A2). Therefore, the modeling of immunotherapy targeting (human) H3.3^{K27M} in rodents is inherently problematic. We addressed this problem by generating mice transgenic for hCD155 and human HLA-A2 (AAD_hCD155-tg mice), and by transducing the mouse CT2A glioma line with H3.3^{K27M} plus an engineered version of human HLA-A2 (CT2A^{AAD_H3.3K27M}) (see our responses to the Editor; items 1. - 5.; pg. 1-3).

Yet, rigorous mechanistic validation of a new immunotherapy modality requires assessment of the response to immune challenge without interference from host-specific factors, such as the epitope:MHC binding in the case of H3.3^{K27M}. For these reasons, we employ the SIINFEKL (OVA) model antigen and the mouse B16F10.9-OVA malignant melanoma model in our mechanistic studies. The B16F10.9-OVA model allows investigating an antigenic epitope in a mouse model without limitations imposed by host MHC-specificity. B16, although not flawless, is arguably the most respected rodent model for cancer immunotherapy research. It is spontaneous, and is regarded as relatively immunologically 'cold' (chemical mutagen-induced murine tumor models are usually tainted by very high mutational burden and unrepresentative intrinsic immune-engagement within the tumor in implanted hosts). Please see our detailed response to comment #7 of Reviewer 1, pg. 12. There is compelling evidence for this in a thorough, systematic study of commonly used mouse tumor models, where B16 was identified as one of the 'coldest', least immune-engaged tumors.¹⁹

Many sophisticated tools available for SIINFEKL (OVA), e.g. OT-I cells (C57Bl6 syngeneic T cells with a SIINFEKL-specific TCR), or H2Kb-SIINFEKL pentamers (Proimmune), have been validated and standardized through countless investigations in many laboratories. We have used these research tools extensively throughout our study, which is indispensable for achieving rigorous mechanistic insight and to evaluate the merits of PVSRIPO vector-based immunization compared to competing strategies.

We therefore, believe that our investigations with mOVA2 *in vitro* and *in vivo* provides most valuable proof-of-concept, mechanistic information about our strategy and its advantages over competing approaches.

However, the Reviewer is correct in insisting on *in vivo* evidence obtained with targeting the H3.3^{K27M} signature of DIPG in an immunocompetent murine orthotopic (intracerebral) malignant glioma model. We have generated such a model, and performed mRIPO(H3.3) immunization studies. See our responses to the **Editor**; items 1. - 5.; pg. 1-3.

“5. Fig. 7F: what are the levels of Granzyme B and IFN-gamma in this co-culture system?”

The Jurkat 76 H3.3^{K27M}_TCR+_CD8+ cells are derived from an engineered Jurkat clone (‘Jurkat 76’) that lacks endogenous TCR and CD8 (both are introduced in the Jurkat 76 H3.3^{K27M}_TCR+_CD8+ cells used in our study, which were obtained from our collaborator, Dr. H. Okada).¹

Jurkat 76 H3.3^{K27M}_TCR+_CD8+ cells produce only IL-2 when stimulated with antigen (personal communication from the team of Dr. H. Okada, who generated these cells). In prior experimentation with the Jurkat 76 line, e.g. Jurkat 76 manipulated to express a TCR specific for a HCV antigenic signature³⁷ or the H3.3^{K27M}_TCR+_CD8+ cells used in the present study,¹ only antigen-induced IL-2 production has been documented.

Furthermore, we employed the Jurkat76 H3.3^{K27M}_TCR+_CD8+ cells merely to demonstrate that RIPO(H3.3)-infected DCs present the H3.3^{K27M} epitope to T cells. They are not used in assays testing T cell function.

“6. Is it possible to set up a co-culture system of human BMDC, CD8+ T cells and HLA-A2+ DIPG cells to demonstrate the efficacy of RIPO(H3.3)?”

This is a very good suggestion and this would be feasible in principle; also, we have experience with such assays.³

Setting up such sophisticated co-culture experiments is not trivial. Given the abundance of empirical evidence for (i) dendritic cell infection, proinflammatory activation, and maturation marker induction by PVSRIPO vectors *in vitro*; (ii) antigen presentation to T cells *in vitro* and *in vivo*; (iii) CD8 T cell co-stimulation *in vitro* and *in vivo*; (iv) effector CD8 T cell responses elicited *in vivo*; and (v) antitumor effects in mouse tumor models elicited by PVSRIPO-based vector immunization *in vivo*, we feel that such *in vitro* co-culture experiments would not substantially add to the body of evidence already available.

“7. A T cell migration assay should be performed with conditioned medium collected from RIPO(H3.3)-infected DCs to demonstrate effective T cell recruitment.”

We have provided compelling evidence for T cell recruitment to the intramuscular inoculation site (**Figure 4b**) and to syngeneic tumors (**Figure 6c**) *in vivo*. We believe that this *in vivo* evidence trumps *in vitro* migration assays, such as T cell migration tests with conditioned media.

Furthermore, we have compiled strong empirical evidence that the most abundant cytokine released upon PVSRIPO infection in mouse bone marrow-derived dendritic cells (**Figure 3c, d**), in skeletal muscle tissue of mOVA2-immunized mice (**Figure 4a**) and in human PBMC-derived dendritic cells (**Figure 7d**) is CXCL10. A ligand for CXCR3 on T cells, CXCL10 was shown to be essential for T cell recruitment to tumors.³⁸

“8. It would be nice if the virology terms (e.g. the processing of VP0, VP3 and VP1 and P2/2BC/2C) are explained for readers not in this field.”

The reviewer is right. We have added a complete polyprotein map, showing the proteolytic cleavages in the viral polyprotein and explaining the identity of the various viral products referenced in the manuscript for the readers' orientation (revised **Figure 2a**).

“9. Sup Fig.1: please show experimental data demonstrating the viability of each vector.”

We added images depicting the plaque phenotype, demonstrating viability, for each construct as applicable (revised **Supplementary Figure 4**).

Minor comments:

The statement “Unlike adult GBM, which is extremely heterogeneous, ~80% of Diffuse Intrinsic Pontine Gliomas (DIPG) and ~20% of pediatric GBMs carry a homogenously expressed driver mutation in histone 3.3 [H3.3K27M]12, 13” is neither complete nor entirely accurate. While it is true that 80% of DIPGs express H3K27M mutation, hemispheric pediatric GBMs express a different histone mutation in ~20% cases, H3G34R/V. Furthermore, other midline gliomas such as thalamic and spinal cord, also express the H3K27M mutation. This entity is now classified as H3K27M+ diffuse midline glioma (DMG) by the WHO and this formal reclassification would be referenced.

The reviewer is right. We have amended our text accordingly (see **revised Introduction**, pg. 4).

Reviewer #4, expert in preclinical development of oncolytic virus (Remarks to the Author):

This is a clearly written and presented manuscript that argues that a picornavirus vector created by the authors can be used as a T cell vaccine specifically for use in cancer indications. The authors argue that while in general picornavirus vectors are not useful for expressing transgenes due to inherent instability, their design of incorporating a peptide sequence into a critical regulatory region of the virus will favour retention of transgenic sequences. This is a strategy that this same group published on in 2003 (Dobrikova et al). They present very nice data that shows their clinically validated vector can stimulate respectable T cell responses against encoded peptides.

Critique

“(1) the authors do not really provide any detailed discussion comparing and contrasting the results from this paper and their 2003 work. Is this an incremental improvement compared to what they have done previously or a significant improvement and why?”

The reviewer raises a valid question. The basic principle underpinning our vector design, functional IRES inserts incentivizing insert retention, was first described in a 2003 manuscript from our group.³⁹ While the 2003 paper introduced the concept of the ‘functional IRES insert’, using model inserts without clinical translational merit, the present study is a substantial advance providing fundamental new insight and establishing a rigorous preclinical basis for future clinical investigations:

- (i) the RIPO-based vector principle introduced in Dobrikova *et al.* (2003) was a basic, proof-of-principle study into the opportunities (in terms of genetic stability) and limitations (in terms of insert size) of IRES recombination. Yet, it was never subjected to tests of immunogenicity *in vitro* or in immunization assays *in vivo*. Our current study is a focused clinical translational study, providing a rational basis for PVSRIPO-based immunization against a clinically valid target (H3.3^{K27M}) including thorough assessment of immunogenic properties *in vitro* and preclinical assays in rodent tumor models *in vivo*.
- (ii) in our new study, we use more sophisticated, validated means to establish genetic stability [in Dobrikova *et al.* (2003), genetic stability of RIPO-based vectors was tested by PCR only, as these early investigations focused primarily on insert retention and the limitations of insert size].³⁹

Genetic stability is a defining regulatory concern with our strategy and the methods used in Dobrikova *et al.* would not be appropriate for establishing a rigorous approach towards optimizing and assessing genetic stability at the level required for an IND.

- (iii) at the time Dobrikova *et al.* (2003) was published, the pivotal roles of macrophages/dendritic cells as natural, high priority targets of poliovirus in

primates had not yet been realized. This only occurred after a landmark study of (wild type) poliovirus infection of non-human primates by the oral route, and careful investigation of virus biodistribution in the gastrointestinal tract was published.⁷ Therefore, a fundamental aspect of our approach and part of the basic premise of our work, PVSRIPO's unusual host relationship manifest in antigen-presenting cells, had not been conceived.

- (iv) at the time Dobrikova *et al.* (2003) was published, the safety of PVSRIPO in rigorous, IND-directed non-human primate toxicology studies, or in human subjects had not been established. IND-directed primate toxicology for PVSRIPO was published in 2012⁸ and clinical trials with PVSRIPO occurred much later (from 2012 onward). We recently reported on PVSRIPO safety in human subjects in 2018.¹¹

Thus, critical safety information underpinning the concept of PVSRIPO-based vector immunization, e.g. in pediatric patients with recurrent DIPG, has become available only recently.

- (v) critical empirical evidence regarding the genetic and functional basis of HRV2 IRES-mediated neuro-attenuation were established only after 2003.^{9, 10, 27, 30, 31, 40} Therefore, the impact of IRES recombination/foreign sequence insertion on PVSRIPO's neuro-attenuation phenotype could not be rationally evaluated at the time Dobrikova *et al.* (2003).

“(2) The observation that the SINNF EKL peptide encoding sequence evolved over time suggests that there may be significant restrictions on what can be encoded...this should be discussed.”

The reviewer is correct. Enterovirus' RNA-dependent RNA polymerases are characterized by one of the highest degrees of infidelity in nature. Thus, genetic instability is intrinsic in any approach based on enterovirus constructs. Because genetic stability of PVSRIPO-based vectors can only be defined functionally (e.g. through serial propagation *in vitro*), but not rationally, it is critical to monitor insert integrity through rigorous empirical methods.

This is the reason why we extensively focused on validating methods to assure epitope retention and expression with the vectors discussed in our study, such as mOVA2 (SIINFEKL) and RIPO(H3.3) [histone 3.3 (K27M)]. Please also consider the detailed studies of initiation codon use we carried out (see revised **Supplementary Figure 4**).

However, while there is a need to thoroughly monitor genetic stability with our approach, this does not impose limits on the type of insert that can be generated, stably inserted or expressed *in vivo*. To show the diversity of sequences suitable for insertion into our platform, we have now included additional vector designs in **new Supplementary Figure 3**, which feature PVSRIPO-based vector prototypes delivering a peptide representing the IDH2(R172G) mutation of malignant gliomas, or a series of SIV p55 Gag-derived fragments.

These vectors were confirmed to be genetically stable with the empirical approach outlined in our study.

We have covered the issue of genetic stability with our vector design in our manuscript at great length; also, the more generic issue of genetic stability of the neuro-attenuation phenotype of PVSRIPO has been thoroughly investigated previously.^{9,27} See also our response to a similar comment from **Reviewer 2**, pg. 17-18.

“(3) the virus cannot encode more than one peptide? Is this the case? These types of limitations of the technology should be discussed.”

The reviewer is correct. As genetic integrity of recombinant vectors is a defining regulatory consideration when assessing PVSRIPO-based vector prototypes, the main limitation of our technology is insert size. Any attempt of generating enterovirus-based vectors where inserts increase the size of the viral genome or add a replicative burden, is doomed to fail, because it will trigger rapid deletion events.

We are exceedingly clear about this fact, because it defines our approach. This is prominently discussed in our study and reverberates with earlier findings in our group.³⁹

“(4) The authors present data that shows about 0.5% of T cells recognize the encoded peptide after PVSRIPO immunization. Can this be boosted with more administrations. Do the authors feel this is a clinically relevant level of T cell stimulation and if so why?”

First, in our assays, CD8 T cell responses were measured by pentamer/tetramer in blood, but antigen-specific CD8 T cells may accumulate in sites where they encounter the target antigen, e.g. in H3.3^{K27M} positive tumors. Second, we measured CD8 T cell responses to immunization, in the absence of constitutive epitope availability. The frequency of antigen-specific CD8 T cells can change dramatically in the presence of antigen (e.g. within a H3.3^{K27M}-positive tumor), and such a frequency increase may occur locally. Third, antigen-specific CD8 T cell frequencies observed in experimental mouse models cannot be extrapolated to the human situation.

Moreover, and perhaps more importantly, there is compelling, published evidence for clinical efficacy of diverse cancer vaccine approaches, yielding target antigen-specific CD8 T cell frequencies at or below the ~0.5% threshold (in human subjects):

[reported in Ott, P.A. et al. An immunogenic personal neoantigen vaccine for patients with melanoma. *Nature* **547**, 217-221 (2017)]: six melanoma patients were enrolled in this study and received 5 prime and 2 booster vaccinations with synthetic plus poly-ICLC administered subcutaneously.⁴¹ IFN- γ intracellular cytokine staining showed between 0.05 to 1.15% of antigen-specific CD8 T cells after immunization. Four out of the six patients in this study showed no tumor recurrence at 25 months after vaccination. The remaining 2 patients had progressive disease and were treated with anti-PD-1 after which they had complete tumor regression.

[reported in Sahin, U. et al. Personalized RNA mutanome vaccines mobilize poly-specific therapeutic immunity against cancer. *Nature* **547**, 222-226 (2017)]: in a trial with RNA-encoded tumor antigen injected directly in advanced-stage melanoma patients' lymph nodes.⁴² In this study, patients were given up to 12 inoculations. After these 'vaccinations', patients had between 0.04 to 0.84% of tumor antigen-specific CD8 T cells in peripheral blood as determined by multimer staining. After immunization, patients in this study cumulatively had a significantly decreased rate of metastasis. Eight of the thirteen patients immunized remained tumor free (between 12 to 23 months post immunization). The remaining five relapsed, one was then treated with anti-PD1 and had a complete response.

“(5) Most of the population has already been vaccinated against polio, will this impact the ability of this vector to initiate immune responses?”

Please see our detailed response to comments by **Reviewer #1** about the issue of preexisting immunity to poliovirus (**comment #4**, pg. 10).

References

1. Chheda, Z.S. et al. Novel and shared neoantigen derived from histone 3 variant H3.3K27M mutation for glioma T cell therapy. *J Exp Med* **215**, 141-157 (2018).
2. Newberg, M.H. et al. Importance of MHC class 1 alpha2 and alpha3 domains in the recognition of self and non-self MHC molecules. *J Immunol* **156**, 2473-2480 (1996).
3. Brown, M.C. et al. Cancer immunotherapy with recombinant poliovirus induces IFN-dominant activation of dendritic cells and tumor antigen-specific CTLs. *Sci Transl Med* **9** (2017).
4. Jahan, N., Wimmer, E. & Mueller, S. A host-specific, temperature-sensitive translation defect determines the attenuation phenotype of a human rhinovirus/poliovirus chimera, PV1(RIPO). *J Virol* **85**, 7225-7235 (2011).
5. Jahan, N., Wimmer, E. & Mueller, S. Polypyrimidine tract binding protein-1 (PTB1) is a determinant of the tissue and host tropism of a human rhinovirus/poliovirus chimera PV1(RIPO). *PloS one* **8**, e60791 (2013).
6. Gromeier, M. & Wimmer, E. Mechanism of injury-provoked poliomyelitis. *J Virol* **72**, 5056-5060 (1998).
7. Shen, L. et al. Pathogenic Events in a Nonhuman Primate Model of Oral Poliovirus Infection Leading to Paralytic Poliomyelitis. *J Virol* **91** (2017).
8. Dobrikova, E.Y. et al. Attenuation of neurovirulence, biodistribution, and shedding of a poliovirus:rhinovirus chimera after intrathalamic inoculation in *Macaca fascicularis*. *J Virol* **86**, 2750-2759 (2012).
9. Campbell, S.A., Lin, J., Dobrikova, E.Y. & Gromeier, M. Genetic determinants of cell type-specific poliovirus propagation in HEK 293 cells. *J Virol* **79**, 6281-6290 (2005).
10. Yang, X. et al. Evaluation of IRES-mediated, cell-type-specific cytotoxicity of poliovirus using a colorimetric cell proliferation assay. *J Virol Methods* **155**, 44-54 (2009).
11. Desjardins, A. et al. Recurrent Glioblastoma Treated with Recombinant Poliovirus. *N Engl J Med* **379**, 150-161 (2018).
12. Wahid, R., Cannon, M.J. & Chow, M. Dendritic cells and macrophages are productively infected by poliovirus. *J Virol* **79**, 401-409 (2005).
13. Diamond, M.S. et al. Type I interferon is selectively required by dendritic cells for immune rejection of tumors. *J Exp Med* **208**, 1989-2003 (2011).
14. Andino, R. et al. Engineering poliovirus as a vaccine vector for the expression of diverse antigens. *Science* **265**, 1448-1451 (1994).
15. Mandl, S., Hix, L. & Andino, R. Preexisting immunity to poliovirus does not impair the efficacy of recombinant poliovirus vaccine vectors. *J Virol* **75**, 622-627 (2001).
16. Hu, S.L. et al. Protection of macaques against SIV infection by subunit vaccines of SIV envelope glycoprotein gp160. *Science* **255**, 456-459 (1992).
17. Kato, H. et al. Differential roles of MDA5 and RIG-I helicases in the recognition of RNA viruses. *Nature* **441**, 101-105 (2006).
18. Ahmad, S. et al. Breaching Self-Tolerance to Alu Duplex RNA Underlies MDA5-Mediated Inflammation. *Cell* **172**, 797-810 e713 (2018).
19. Lechner, M.G. et al. Immunogenicity of murine solid tumor models as a defining feature of in vivo behavior and response to immunotherapy. *J Immunother* **36**, 477-489 (2013).

20. Mitchell, D.A., Fecci, P.E. & Sampson, J.H. Immunotherapy of malignant brain tumors. *Immunol Rev* **222**, 70-100 (2008).
21. Wilson, E.H., Weninger, W. & Hunter, C.A. Trafficking of immune cells in the central nervous system. *J Clin Invest* **120**, 1368-1379 (2010).
22. Wille-Reece, U., Wu, C.Y., Flynn, B.J., Kedl, R.M. & Seder, R.A. Immunization with HIV-1 Gag protein conjugated to a TLR7/8 agonist results in the generation of HIV-1 Gag-specific Th1 and CD8+ T cell responses. *Journal of immunology* **174**, 7676-7683 (2005).
23. Mangani, D. et al. Limited role for transforming growth factor-beta pathway activation-mediated escape from VEGF inhibition in murine glioma models. *Neuro Oncol* **18**, 1610-1621 (2016).
24. Reardon, D.A. et al. Glioblastoma Eradication Following Immune Checkpoint Blockade in an Orthotopic, Immunocompetent Model. *Cancer Immunol Res* **4**, 124-135 (2016).
25. Schreiber, K., Rowley, D.A., Riethmuller, G. & Schreiber, H. Cancer immunotherapy and preclinical studies: why we are not wasting our time with animal experiments. *Hematol Oncol Clin North Am* **20**, 567-584 (2006).
26. Kew, O.M., Sutter, R.W., de Gourville, E.M., Dowdle, W.R. & Pallansch, M.A. Vaccine-derived polioviruses and the endgame strategy for global polio eradication. *Annu Rev Microbiol* **59**, 587-635 (2005).
27. Dobrikova, E.Y. et al. Recombinant oncolytic poliovirus eliminates glioma in vivo without genetic adaptation to a pathogenic phenotype. *Mol Ther* **16**, 1865-1872 (2008).
28. Gromeier, M., Alexander, L. & Wimmer, E. Internal ribosomal entry site substitution eliminates neurovirulence in intergeneric poliovirus recombinants. *Proc Natl Acad Sci U S A* **93**, 2370-2375 (1996).
29. Gromeier, M., Bossert, B., Arita, M., Nomoto, A. & Wimmer, E. Dual stem loops within the poliovirus internal ribosomal entry site control neurovirulence. *J Virol* **73**, 958-964 (1999).
30. Merrill, M.K., Dobrikova, E.Y. & Gromeier, M. Cell-type-specific repression of internal ribosome entry site activity by double-stranded RNA-binding protein 76. *J Virol* **80**, 3147-3156 (2006).
31. Merrill, M.K. & Gromeier, M. The double-stranded RNA binding protein 76:NF45 heterodimer inhibits translation initiation at the rhinovirus type 2 internal ribosome entry site. *J Virol* **80**, 6936-6942 (2006).
32. Brown, M.C. & Gromeier, M. MNK Controls mTORC1:Substrate Association through Regulation of TELO2 Binding with mTORC1. *Cell Rep* **18**, 1444-1457 (2017).
33. Dobrikov, M.I., Dobrikova, E.Y. & Gromeier, M. Dynamic Regulation of the Translation Initiation Helicase Complex by Mitogenic Signal Transduction to Eukaryotic Translation Initiation Factor 4G. *Mol Cell Biol* **33**, 937-946 (2013).
34. Dobrikov, M.I., Dobrikova, E.Y. & Gromeier, M. Ribosomal RACK1:PKCbeta11 modulates intramolecular interactions between unstructured regions of eIF4G that control eIF4E and eIF3 binding. *Mol Cell Biol* (2018).
35. Dobrikov, M.I., Dobrikova, E.Y. & Gromeier, M. Ribosomal RACK1:PKCbeta11 phosphorylates eIF4G1(S1093) to modulate cap-dependent and -independent translation initiation. *Mol Cell Biol* (2018).

36. Dobrikov, M.I., Shveygert, M., Brown, M.C. & Gromeier, M. Mitotic phosphorylation of eukaryotic initiation factor 4G1 (eIF4G1) at Ser1232 by Cdk1:cyclin B inhibits eIF4A helicase complex binding with RNA. *Mol Cell Biol* **34**, 439-451 (2014).
37. Zhang, Y. et al. Transduction of human T cells with a novel T-cell receptor confers anti-HCV reactivity. *PLoS pathogens* **6**, e1001018 (2010).
38. Mikucki, M.E. et al. Non-redundant requirement for CXCR3 signalling during tumoricidal T-cell trafficking across tumour vascular checkpoints. *Nat Commun* **6**, 7458 (2015).
39. Dobrikova, E.Y., Florez, P. & Gromeier, M. Structural determinants of insert retention of poliovirus expression vectors with recombinant IRES elements. *Virology* **311**, 241-253 (2003).
40. Goetz, C., Everson, R.G., Zhang, L.C. & Gromeier, M. MAPK signal-integrating kinase controls cap-independent translation and cell type-specific cytotoxicity of an oncolytic poliovirus. *Mol Ther* **18**, 1937-1946 (2010).
41. Ott, P.A. et al. An immunogenic personal neoantigen vaccine for patients with melanoma. *Nature* **547**, 217-221 (2017).
42. Sahin, U. et al. Personalized RNA mutanome vaccines mobilize poly-specific therapeutic immunity against cancer. *Nature* **547**, 222-226 (2017).

Reviewers' comments:

Reviewer #1 (Remarks to the Author):

Mosaheb et al. have provided a revised version of their manuscript. The revisions, experiments and explanations have been extensive and they should be commended for the tremendous amount of work and thought that went into this. The most relevant revision is that they now have generated the tools and reagents needed to address their hypothesis. Although title and abstract focus on the technical aspects of having engineered a strategy to express foreign antigens in polio, the proof-of-principle experiments are based on the Ova model and then on physiologically relevant H3.3 K27M antigen expressed in DIPG.

The technical aspects related to stable expression of peptide antigens (figures 1 and 2) are better explained and more robust. Figure 3, Figure 4a-c, Figure 6 show the immune response in the Ova model. The more physiologic relevant experiments are in Figure 4d, 5e, f, figure 7, 8 which show the transgenic mouse model and the immune experiments related to this model. These experiments clearly make this paper more robust than the previous version.

My major comments are:

1- The back and forth between OVA and H3.3 is very confusing. Since the most relevant part here is H3.3 (the physiologically relevant target), please consider placing the less interesting OVA experiments in the supplementary data

2- Figure 2e: the asterisks are confusing and the band that is meant to represent H3.3. are also not clear. I understand that there is time-related processing but it takes quite some time for a reader to look at these blots and try to figure which bands are H3.3

3- All in vitro experiments are conducted at a MOI of 10. IS this in vitro dose physiologically relevant to what is administered in vivo?

4- Figure 4d- This is important data related to immunization. A lymph node with H3.3 positive IHC is shown. The authors say in the legend that these are APCs but there is no evidence of this. Staining for CD11c is also shown but it is not clear if these are the same cells as the H3.3 positive cells. There is also no quantitative analyses of number of cells that are H3.3 positive and number of draining Lymph Nodes assayed when compared to controls of SUPpl. Figure 11, 12.

5- The Authors propose the exciting concept that polio's effects are that of a vaccine more than a cytotoxic virus. The in vitro and in vivo comparisons of Figure 5a-e between different vaccination strategies are thus important to show polio superiority in immune assays. But then the true test of whether the polio vaccination strategy is superior would have been a head- to-head comparison in mice with tumors. But figure 6a and 7e only show mOVA2 and mRIPOH3.3, respectively against mRIPOdelta6. The more important controls of poly(I:C) + peptide or mRIPOdelta6 + peptide are now shown. I think this is important data and controls because the difference in tumor activity between mRIPOH3.3. mRIPO OVA2 and RIPODelta6 may be significant but is small and Figure 5c, d show that the peptides alone + poly(I:C) may have more T cell activation systemically compared to mRIPdelta6 alone

Reviewer #2 (Remarks to the Author):

The authors have improved the quality of the manuscript. In addition to the in vitro data from the previous version, they developed an in vivo model to address the questions that were asked by the different reviewers.

In terms of novelty they argue that their virus can be differentiated from others, raising points in terms of stimulating antigen presentation and T-cell co-stimulation instead of repressing as might occur with other DNA viruses.

They do present mechanistic data to support their claims that PVSRIPO-based vectors achieve mainly 3 things: 1) antigen expression, 2) type I/III-dominant pro-inflammatory signaling stimulation, and 3) maturation marker induction in infected dendritic cells.

They have presented new data in this submission, including specifically for this vector to be used as a platform to express any foreign transgene antigen sequence. This may allow the use of this vector for the vaccination against other tumor types in humans, and hence this may be more generalizable than previously noted. The authors have not generated data from new animal models for each antigen that they propose to test.

Reviewer #3 (Remarks to the Author):

The manuscript is improved and the authors have answered many of my concerns. However, the writing still needs work. The authors should endeavor to make this easier to read, trying wherever possible to avoid "alphabet soup".

Reviewer #4 (Remarks to the Author):

The authors have gone to great lengths to address all of the reviewers concerns including a considerable amount of new data. I think they provide excellent and well controlled experiments. While I agree that the immune responses they are seeing in their mouse experiments are comparable to immune responses seen in clinical studies (e.g. they cite the work from Ugur Sahin and colleagues), they are assessing responses in mice (as they acknowledge) which in general notoriously over-estimate responses that are actually found in humans. None-the-less an interesting application of their technology and I believe it merits testing in the clinical setting to understand its full potential. John Bell

Reviewer #1 (Remarks to the Author):

Mosaheb et al. have provided a revised version of their manuscript. The revisions, experiments and explanations have been extensive and they should be commended for the tremendous amount of work and thought that went into this. The most relevant revision is that they now have generated the tools and reagents needed to address their hypothesis. Although title and abstract focus on the technical aspects of having engineered a strategy to express foreign antigens in polio, the proof-of-principle experiments are based on the Ova model and then on physiologically relevant H3.3 K27M antigen expressed in DIPG. The technical aspects related to stable expression of peptide antigens (figures 1 and 2) are better explained and more robust. Figure 3, Figure 4a-c, Figure 6 show the immune response in the Ova model. The more physiologic relevant experiments are in Figure 4d, 5e, f, figure 7, 8 which show the transgenic mouse model and the immune experiments related to this model. These experiments clearly make this paper more robust than the previous version.

My major comments are:

1- The back and forth between OVA and H3.3 is very confusing. Since the most relevant part here is H3.3 (the physiologically relevant target), please consider placing the less interesting OVA experiments in the supplementary data

We have followed this suggestion and moved non-essential information relating to the mOVA2 vector to the Supplement (see **comment #2**, pg. 5-6).

However, we feel that principal OVA data, reported in Figs. 3-6, *must* remain in the main body of the manuscript. We describe a new technology, slated for first-in-man investigation, with claims of mechanistic superiority over conventional approaches. Empirical rigor demands that our claims are supported by investigations in standardized models with validated epitopes, without interference from host-, or target epitope-specific variables. We used rigorous assays based on standardized OVA tools (e.g. OT-1 CD8 T cells, H2Kb-SIINFEKL pentamer, etc.), which will enable readers to better gauge the merits of our technology.

H3.3^{K27M} is not a mouse epitope and the AAD/CD155 double-transgenic mouse model we must use to enable presentation of an HLA-A2 epitope in mice is somewhat contrived (it has an artificial, chimeric mouse-human MHC). Because of its spontaneous origin and relatively low intrinsic immunogenicity, B16F10 arguably is the most respected immunocompetent rodent tumor model. For example, it was instrumental in establishing the paradigm-shift of immune checkpoint blockade.¹⁶ mOVA2 immunization experiments against B16F10-OVA involve the endogenous mouse MHC I (H2Kb) and demonstrate the mechanisms of our vector approach without interference from host-specific confounders.

2- Figure 2e: the asterisks are confusing and the band that is meant to represent H3.3. are also not clear. I understand that there is time-related processing but it takes quite some time for a reader to look at these blots and try to figure which bands are H3.3

The reviewer is right, the figure was difficult to read. The complicated depiction was a result of including data of divergent assays (immunoprecipitation followed by immunoblot, vs. 'direct' immunoblot), involving two different vector constructs, in the same figure.

We now moved the description of foreign epitope expression with mOVA2 to the **new Supplementary Figure 5**. This yielded a much simpler, easier to read **revised Figure 2** in the main manuscript, which exclusively focuses on RIPO(H3.3) and the expression of the H3.3^{K27M} epitope. We now clearly labeled the viral fusion-polypeptides including the H3.3^{K27M} signature (see **revised Figure 2b**).

3- All in vitro experiments are conducted at a MOI of 10. IS this in vitro dose physiologically relevant to what is administered in vivo?

This is a valid point. Poliovirus infections ***in vitro*** are naturally inefficient, as 90% of input virus is 'sloughed off' from cells upon CD155 receptor interaction as 135S particles that do not enter host cells.¹⁷ Thus, *in vitro*, only ~10% of input poliovirus successfully completes the entry process and initiates infection. Therefore, the 'actual' MOI with poliovirus is far lower than the input virus.

It is unknown if such sloughing off occurs *in vivo*.

To account for the sloughing phenomenon, *in vitro* studies with poliovirus classically are conducted with MOIs of 10, as to reach a 'true' MOI of 1 with the objective to target every cell in the culture. Synchronized infection at an MOI of 10 (corresponding to a 'true' MOI of 1) has been the mainstay of poliovirus research for >50 years, as it provides a rigorous empirical framework for deciphering virus:host relations during the infection process.

Our studies are geared to achieve fundamental mechanistic insight into virus/vector:host interactions in human and mouse dendritic cells. Infections at MOIs <10 would yield samples where only a subset of cells are infected. This would obfuscate the many immunoblot and flow cytometry assays we have conducted and greatly complicate the interpretation of our results.

Based on the experience with PVSRIPO in human subjects,¹⁸ we anticipate a clinical dose in the range of $\sim 10^7$ - 10^8 TCID for intramuscular administration of RIPO(H3.3). An empirical MOI of 10 in our mechanistic *in vitro* studies is consistent with this anticipated clinical dose range.

We have previously characterized the DC phenotype of the vector parent (PVSRIPO) at an MOI of 1 and observed similar sub-lethal infection with type I interferon-dominant stimulation as the data reported in the present study.¹⁹

We also point the reviewer to the *in vivo* assays of PVSRIPO vector DC infection in our report (see **Figure 4**). I.M. immunization leads to loco-regional induction of CCL2, CCL5, CXCL1 and CXCL10 *in vivo* (**Figure 4a**). CCL2 (MCP-1) and CCL5 (RANTES) are potent DC chemokines,^{20, 21} inducing their migration to the immunization site. Accordingly, flow

cytometry analyses of the immunization site demonstrated an influx of CD11c+ and CD11b+ cells, suggesting migration of antigen presenting cells (DCs) to the site of immunization (**Figure 4b**). Once they migrate to the immunization site, such DCs would become targets for PVSRIPO vector infection. This is indeed reflected in our studies, as locoregional DCs induced the CD40/CD86 maturation markers (**Figure 4c**). Therefore, hallmarks of PVSRIPO vector infection of DCs at an MOI of 10 *in vitro*, also were observed *in vivo*.

4- Figure 4d- This is important data related to immunization. A lymph node with H3.3 positive IHC is shown. The authors say in the legend that these are APCs but there is no evidence of this. Staining for CD11c is also shown but it is not clear if these are the same cells as the H3.3 positive cells. There is also no quantitative analyses of number of cells that are H3.3 positive and number of draining Lymph Nodes assayed when compared to controls of SUppl. Figure 11, 12.

The reviewer is correct. Mouse popliteal lymph nodes are minuscule – we were not able to collect consecutive sections (7-micron thickness) showing the same germinal center (containing the H3.3^{K27M}+ cells) for CD11c co-staining.

To account for the Reviewer's concern, we removed the CD11c staining image because we cannot demonstrate co-staining for H3.3^{K27M} and CD11c on the same cells (see **revised Figure 4**). We also tempered our description and simply state that 'RIPO(H3.3)-mediated H3.3^{K27M} expression occurs in the draining lymph node' (see **revised Results**, pg. 11). We do not claim that H3.3^{K27M} positive cells are DCs in our paper.

Our descriptive immunohistochemical (IHC) analyses detecting H3.3^{K27M}-positive cells in popliteal lymph nodes after i.m. immunization, do not provide definitive evidence that such cells (presumably antigen presenting cells) are driving H3.3^{K27M}-directed CD8 T cell responses. However, our IHC results illustrate that RIPO(H3.3)-infected cells drain to local lymph nodes and, thus, corroborate important mechanistic *in vitro* analyses. For example, RIPO(H3.3) infection of human DCs *in vitro* strongly induced CCR7 (Fig. 8c), a mediator of lymph node migration in DCs.

Staining of mock lymph nodes (Supplementary Fig. 12) and isotype control staining (Supplementary Fig. 13) show that the H3.3^{K27M} IHC staining in lymph nodes of RIPO(H3.3)-immunized mice is specific. Regarding the number of cells in lymph nodes expressing H3.3^{K27M}: lymph nodes consist primarily of T and B cells that poliovirus cannot infect and, thus, should not express the H3.3^{K27M} epitope. DCs are extremely rare *in vivo*; peripheral DCs migrate to the draining lymph node after activation [e.g. after infection with mRIPO(H3.3)] and CCR7 upregulation (Fig. 8c) to present antigen to T cells (please see comprehensive reviews for detail^{22, 23}). Because DCs are exceedingly rare *in vivo*, the distribution and extent of H3.3^{K27M} staining in a 7-micron section of a mouse popliteal lymph node is expected. While DCs are rare, they are extremely potent inducers of T cell responses; a single DC can stimulate thousands of T cells^{24, 25}. It has been estimated that only 85 antigen-presenting DCs are required to elicit a T cell response in humans²⁴.

DC-mediated CD8 T cell responses are not necessarily dependent on the frequency of activated, epitope-presenting DCs in lymph nodes, but are contingent on the quality of proinflammatory stimulation and activation provided by the context of antigen uptake and presentation. We have compelling evidence buttressed by multiple state-of-the-art complimentary assays, that PVSRIPO vectored epitope delivery induces antigen-specific CD8 T cell responses. Therefore, we have not endeavored to quantify the presence of H3.3^{K27M}-positive cells in lymph nodes.

5- The Authors propose the exciting concept that polio's effects are that of a vaccine more than a cytotoxic virus. The in vitro and in vivo comparisons of Figure 5a-e between different vaccination strategies are thus important to show polio superiority in immune assays. But then the true test of whether the polio vaccination strategy is superior would have been a head- to-head comparison in mice with tumors. But figure 6a and 7e only show mOVA2 and mRIPOH3.3, respectively against mRIPDelta6. The more important controls of poly(i:C) + peptide or mRIPDelta6 + peptide are now shown. I think this is important data and controls because the difference in tumor activity between mRIPOH3.3. mRIP OVA2 and RIPDelta6 may be significant but is small and Figure 5c, d show that the peptides alone + poly(I:C) may have more T cell activation systemically compared to mRIPdelta6 alone

While comparison of our PVSRIPO vector approach with conventional peptide + adjuvant regimens in rodent tumor models may appear sensible, such assays are not feasible or advisable for two main reasons:

First, such efforts are rendered almost impossible due to the dizzying variety of adjuvant regimens and administration methods/schedules that are being employed in conjunction with peptide immunization in the clinic. To illustrate this point, we provide details on the adjuvants used for recent peptide immunization trials in malignant glioma patients:

1. NCT02960230, peptide vaccine targeting H3.3^{K27M} in DIPG: H3.3^{K27M} peptide vaccination **combined with tetanus toxoid peptide emulsified in Montanide** plus **poly(IC-LC)**.
2. NCT02454634, peptide vaccine targeting IDH1^{R132H} in malignant glioma: peptide is **emulsified in Montanide**, administered subcutaneously, and **imiquimod** is given topically.
3. NCT02193347, IDH1^{R132H} peptide vaccine in malignant glioma: peptide, **combined with GMCSF, Montanide, Tetanus toxoid** and is administered intradermally.
4. NCT00643097, EGFRviii peptide vaccine targeting glioblastoma: peptide **conjugated to KLH** and administered **with GMCSF** intradermally.
5. NCT03299309, CMV Peptide vaccine targeting glioblastoma: patients are first preconditioned intradermally **with tetanus toxoid** 6 to 24h prior to vaccination with **peptide in Montanide**.

Since there is no agreement on prioritizing adjuvant strategies and there are no benchmark immune monitoring standards guiding their use, it is impossible to define the correct regimen to which PVSRIPO should be compared. We often use high molecular weight poly(I:C) in our studies as a mechanistic comparator to PVSRIPO, because it resembles viral dsRNA as an innate immune trigger.

Second, the record of rodent tumor models in predicting immunotherapy efficacy in the clinic is dismal (see our detailed response to **comment #10** of **Reviewer 1** in the prior rebuttal letter). We kindly point out to the reviewer that our stance on the utility of rodent tumor models in predicting clinical outcomes is shared by **Reviewer 4** (see pg. 7-8).

Ample precedent teaches that comparing PVSRIPO to conventional approaches in such models will almost certainly prevent reaching empirical conclusions that hold up in the clinic. A much better approach towards rigorous, robust preclinical analysis are *mechanistic* studies in relevant (human) model systems. This is the purpose of our study.

Reviewer #2 (Remarks to the Author):

The authors have improved the quality of the manuscript. In addition to the in vitro data from the previous version, they developed an in vivo model to address the questions that were asked by the different reviewers. In terms of novelty they argue that their virus can be differentiated from others, raising points in terms of stimulating antigen presentation and T-cell co-stimulation instead of repressing as might occur with other DNA viruses. They do present mechanistic data to support their claims that PVSRIPO-based vectors achieve mainly 3 things: 1) antigen expression, 2) type I/III-dominant pro-inflammatory signaling stimulation, and 3) maturation marker induction in infected dendritic cells. They have presented new data in this submission, including specifically for this vector to be used as a platform to express any foreign transgene antigen sequence. This may allow the use of this vector for the vaccination against other tumor types in humans, and hence this may be more generalizable than previously noted. The authors have not generated data from new animal models for each antigen that they propose to test.

The SIV and IDH2-epitope expressing vectors were included to address a prior review comment that documenting the specific mOVA2 and RIPO(H3.3) constructs did not properly support the claim of generalizability of our vector design (see our response to **Reviewer 2**, pg. 17-18 in the prior rebuttal letter). We added the additional vector prototypes in **Supplementary Figure 3** to demonstrate that our vector design is agnostic to specific IRES insert sequences. These vector prototypes represent proof-of-principle constructs that were evaluated for genetic stability *in vitro* only (we did not propose to advance these designs to animal testing).

Testing the vector prototypes shown in **Supplementary Figure 3** in *in vivo* immunization studies is outside the scope of this report and would be a massive endeavor, since none of the target epitopes are predicted to bind to murine MHC Class I.

Reviewer #3 (Remarks to the Author):

The manuscript is improved and the authors have answered many of my concerns. However, the writing still needs work. The authors should endeavor to make this easier to read, trying wherever possible to avoid "alphabet soup".

We agree and have followed this suggestion. For example, we moved the complicated description of OVA-fusion polypeptide expression in mOVA2-infected cells to the Supplement (**new Supplementary Figure 5**) and streamlined the corresponding section in the main manuscript. The result is a much simpler to follow **new Figure 2** and related text. We also eliminated superfluous detail, jargon, or complexity where ever this was possible (see revised text in the compare-file document).

Reviewer #4 (Remarks to the Author):

The authors have gone to great lengths to address all of the reviewers concerns including a considerable amount of new data. I think they provide excellent and well controlled experiments. While I agree that the immune responses they are seeing in their mouse experiments are comparable to immune responses seen in clinical studies (e.g. they cite the work from Ugur Sahin and colleagues), they are assessing responses in mice (as they acknowledge) which in general notoriously over-estimate responses that are actually found in humans. None-the-less an interesting application of their technology and I believe it merits testing in the clinical setting to understand its full potential. John Bell

1. Mellman, I., Coukos, G. & Dranoff, G. Cancer immunotherapy comes of age. *Nature* **480**, 480-489 (2011).
2. Curtsinger, J.M., Lins, D.C. & Mescher, M.F. Signal 3 determines tolerance versus full activation of naive CD8 T cells: dissociating proliferation and development of effector function. *J Exp Med* **197**, 1141-1151 (2003).
3. Schwartz, R.H. T cell anergy. *Annual review of immunology* **21**, 305-334 (2003).
4. Steinman, R.M., Turley, S., Mellman, I. & Inaba, K. The induction of tolerance by dendritic cells that have captured apoptotic cells. *J Exp Med* **191**, 411-416 (2000).
5. Shen, L. et al. Pathogenic Events in a Nonhuman Primate Model of Oral Poliovirus Infection Leading to Paralytic Poliomyelitis. *J Virol* **91** (2017).
6. Joffre, O.P., Segura, E., Savina, A. & Amigorena, S. Cross-presentation by dendritic cells. *Nat Rev Immunol* **12**, 557-569 (2012).
7. Deng, L. et al. STING-Dependent Cytosolic DNA Sensing Promotes Radiation-Induced Type I Interferon-Dependent Antitumor Immunity in Immunogenic Tumors. *Immunity* **41**, 843-852 (2014).
8. Diamond, M.S. et al. Type I interferon is selectively required by dendritic cells for immune rejection of tumors. *J Exp Med* **208**, 1989-2003 (2011).
9. Kranz, L.M. et al. Systemic RNA delivery to dendritic cells exploits antiviral defence for cancer immunotherapy. *Nature* **534**, 396-401 (2016).
10. Yang, X. et al. Targeting the tumor microenvironment with interferon-beta bridges innate and adaptive immune responses. *Cancer Cell* **25**, 37-48 (2014).

11. Zitvogel, L., Galluzzi, L., Kepp, O., Smyth, M.J. & Kroemer, G. Type I interferons in anticancer immunity. *Nat Rev Immunol* **15**, 405-414 (2015).
12. Sharpe, A.H. Mechanisms of costimulation. *Immunol Rev* **229**, 5-11 (2009).
13. Petersen, J.L., Morris, C.R. & Solheim, J.C. Virus evasion of MHC class I molecule presentation. *J Immunol* **171**, 4473-4478 (2003).
14. Kato, H. et al. Differential roles of MDA5 and RIG-I helicases in the recognition of RNA viruses. *Nature* **441**, 101-105 (2006).
15. Ahmad, S. et al. Breaching Self-Tolerance to Alu Duplex RNA Underlies MDA5-Mediated Inflammation. *Cell* **172**, 797-810 e713 (2018).
16. Curran, M.A., Montalvo, W., Yagita, H. & Allison, J.P. PD-1 and CTLA-4 combination blockade expands infiltrating T cells and reduces regulatory T and myeloid cells within B16 melanoma tumors. *Proceedings of the National Academy of Sciences of the United States of America* **107**, 4275-4280 (2010).
17. Joklik, W.K. & Darnell, J.E., Jr. The adsorption and early fate of purified poliovirus in HeLa cells. *Virology* **13**, 439-447 (1961).
18. Desjardins, A. et al. Recurrent Glioblastoma Treated with Recombinant Poliovirus. *N Engl J Med* **379**, 150-161 (2018).
19. Brown, M.C. et al. Cancer immunotherapy with recombinant poliovirus induces IFN-dominant activation of dendritic cells and tumor antigen-specific CTLs. *Sci Transl Med* **9** (2017).
20. Ma, Y. et al. CCL2/CCR2-dependent recruitment of functional antigen-presenting cells into tumors upon chemotherapy. *Cancer Res* **74**, 436-445 (2014).
21. Sozzani, S. et al. Migration of dendritic cells in response to formyl peptides, C5a, and a distinct set of chemokines. *Journal of immunology* **155**, 3292-3295 (1995).
22. Alvarez, D., Vollmann, E.H. & von Andrian, U.H. Mechanisms and consequences of dendritic cell migration. *Immunity* **29**, 325-342 (2008).
23. Banchereau, J. & Steinman, R.M. Dendritic cells and the control of immunity. *Nature* **392**, 245-252 (1998).
24. Celli, S. et al. How many dendritic cells are required to initiate a T-cell response? *Blood* **120**, 3945-3948 (2012).
25. Steinman, R.M. & Witmer, M.D. Lymphoid dendritic cells are potent stimulators of the primary mixed leukocyte reaction in mice. *Proc Natl Acad Sci U S A* **75**, 5132-5136 (1978).

REVIEWERS' COMMENTS:

Reviewer #1 (Remarks to the Author):

In this re-revised version, the authors went to great length to try and answer my questions. The experiment that I thought was critical they have performed. I thus have no further suggestions

Dear Dr. Bondar,

Thank you for communicating the results of the evaluation of our manuscript. Please see our response to Reviewer #1 below.

REVIEWERS' COMMENTS:

Reviewer #1 (Remarks to the Author):

In this re-revised version, the authors went to great length to try and answer my questions. The experiment that I thought was critical they have performed. I thus have no further suggestions

We thank the reviewer for his many helpful and constructive suggestions.

Sincerely
M. Gromeier, MD
Duke University